# Isotype-aware inference of B cell clonal lineage trees from single-cell sequencing data

## Graphical abstract

## Highlights

- B cells undergo somatic hypermutation (SHM) and class switch recombination (CSR)

- scRNA-seq reveals both heavy- and light-chain BCR sequences and isotype expression

- TRIBAL infers B cell lineage trees that model SHM and CSR from scRNA-seq data

## Authors

Leah L. Weber, Derek Reiman, Mrinmoy S. Roddur, Yuanyuan Qi, Mohammed El-Kebir, Aly A. Khan

## Correspondence

melkebir@illinois.edu (M.E.-K.), aakhan@uchicago.edu (A.A.K.)

## In brief

During adaptive immune response, B cells undergo somatic hypermutation (SHM) and class switch recombination (CSR). Single-cell RNA sequencing reveals paired heavy- and light-chain sequences and isotypes of B cells. Weber et al. present TRIBAL, which uses these data to accurately infer B cell lineage trees and isotype transition probabilities, modeling both SHM and CSR.

Weber et al., 2024, Cell Genomics 4, 100637
September 11, 2024 © 2024 The Author(s). Published by Elsevier Inc.

CellPress

## Technology

# Isotype-aware inference of B cell clonal lineage trees from single-cell sequencing data

Leah L. Weber,[1] Derek Reiman,[2] Mrinmoy S. Roddur,[1] Yuanyuan Qi,[1] Mohammed El-Kebir,[1,5,*] and Aly A. Khan[2,3,4,*]

[1]Department of Computer Science, University of Illinois at Urbana-Champaign, Urbana, IL 61801, USA
[2]Toyota Technological Institute at Chicago, Chicago, IL 60637, USA
[3]Department of Pathology, University of Chicago, Chicago, IL 60637, USA
[4]Chan Zuckerberg Biohub Chicago, Chicago, IL 60642, USA
[5]Lead contact
*Correspondence: melkebir@illinois.edu (M.E.-K.), aakhan@uchicago.edu (A.A.K.)

## SUMMARY

Single-cell RNA sequencing (scRNA-seq) enables comprehensive characterization of the micro-evolutionary processes of B cells during an adaptive immune response, capturing features of somatic hypermutation (SHM) and class switch recombination (CSR). Existing phylogenetic approaches for reconstructing B cell evolution have primarily focused on the SHM process alone. Here, we present tree inference of B cell clonal lineages (TRIBAL), an algorithm designed to optimally reconstruct the evolutionary history of B cell clonal lineages undergoing both SHM and CSR from scRNA-seq data. Through simulations, we demonstrate that TRIBAL produces more comprehensive and accurate B cell lineage trees compared to existing methods. Using real-world datasets, TRIBAL successfully recapitulates expected biological trends in a model affinity maturation system while reconstructing evolutionary histories with more parsimonious class switching than state-of-the-art methods. Thus, TRIBAL significantly improves B cell lineage tracing, useful for modeling vaccine responses, disease progression, and the identification of therapeutic antibodies.

## INTRODUCTION

Single-cell sequencing technologies have emerged as a powerful tool for understanding and modeling cellular evolution.[1–4] These technologies offer a precise method to trace cell lineage through the observation of genomic and somatic changes, allowing for an in-depth study of the role genetic variation plays in determining cell fitness across various environments, including in cancer and immune response.[5–12] Constructing lineage trees from single-cell data poses a significant challenge, primarily due to phylogenetic uncertainty. Phylogenetic uncertainty refers to the lack of confidence or certainty in the inferred evolutionary relationships among sequenced cells and manifests in two key ways. First, current phylogenetic methods often produce multiple plausible phylogenies that equally explain the observed data. Second, due to insufficient data, the inferred lineage trees may contain unresolved evolutionary relationships, represented by polytomies; i.e., multifurcating nodes with more than two children. This is counter to the underlying cell lineage tree, which is bifurcating, as cell division results in exactly two daughter cells. To confidently infer the key relationships between genetic modifications and cellular fitness, lineage tree inference methods from single-cell data should strive to minimize phylogenetic uncertainty.

One common approach to resolve phylogenetic uncertainty is the inclusion of additional data or constraints to the system in question, such as the use of physical location for studying cancer migration and metastasis[13] or geographical location for the inference of gene flow.[14] Here, we focus on B cell lineage inference and propose a novel method that integrates known biological constraints of antibody class switching, as well as measurements of the antibody class of each sequenced cell, to construct B cell lineage trees that accurately trace the evolutionary trajectory of B cells in a single-cell sequencing experiment. B cells play a pivotal role in the adaptive immune response, producing antibodies that neutralize foreign substances and infections.[15,16] Antibodies are initially formed as sequence-specific B cell receptors (BCRs) consisting of a heavy chain and a light chain (Figure 1A). To enhance their effectiveness, B cells undergo affinity maturation (Figure 1B),[17] a micro-evolutionary process involving repeated cycles of somatic hypermutation (SHM) and cellular divisions. SHM introduces mutations in the BCR genes, selecting for B cells expressing high-affinity BCRs while eliminating those with low affinity.

With single-cell RNA sequencing (scRNA-seq), it is now possible to efficiently assemble BCR sequences that include both the heavy and light chain from a population of B cells[18] (Figure 1C). As a result, the evolution of BCRs during affinity maturation can now be traced phylogenetically using scRNA-seq with high fidelity.[4] The selection pressures applied to B cells during the affinity maturation process necessitate more specialized analytical approaches than those utilized for species phylogeny inference.[19–22] Specifically, Hoehn et al. developed HLP17[8] and HLP19,[23] which are specialized codon substitution models for

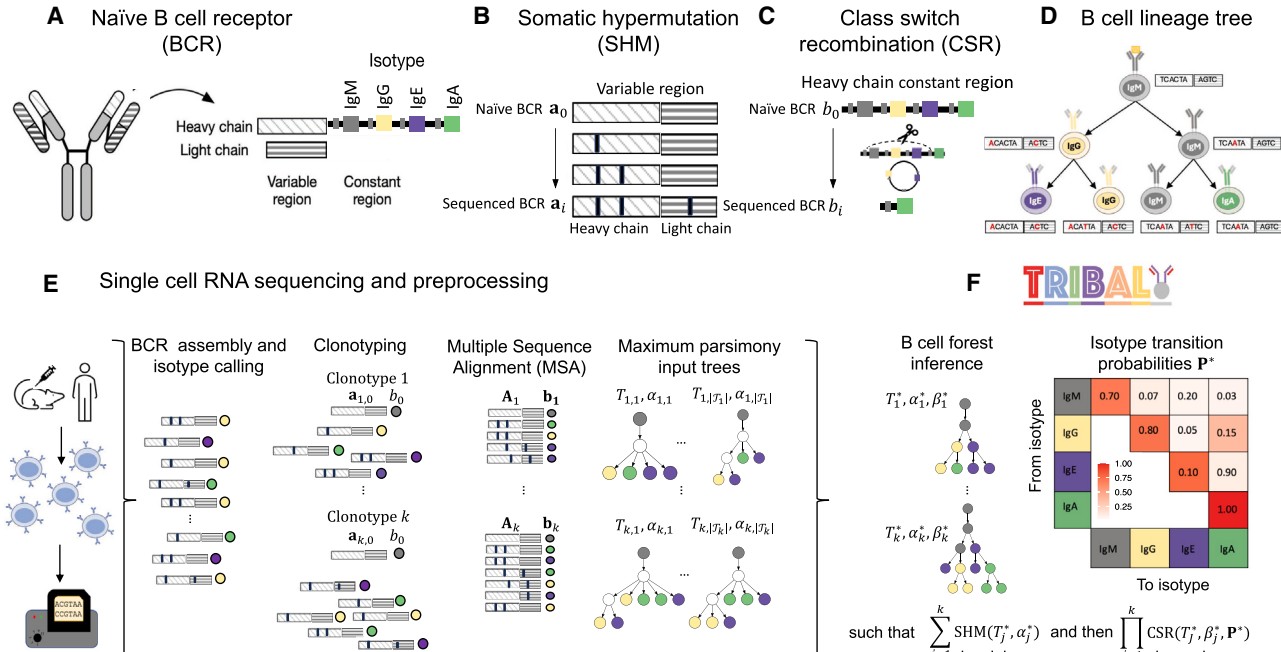

**Figure 1. TRIBAL infers B cell lineage trees and isotype transition probabilities for scRNA-seq data**

(A) A BCR consists of paired heavy and light immunoglobulin chains, each consisting of a variable and constant region. The isotype is the heavy-chain constant locus that is transcribed.

(B) The BCR undergoes SHM/affinity maturation, where point mutations are introduced into the variable region of the heavy and light chains.

(C) B cells also undergo CSR, where the heavy-chain constant locus undergoes recombination and begins transcribing a different isotype.

(D) These two processes can be modeled with a B cell lineage tree that captures the evolutionary relationships between B cells as well as the sequences and isotypes of ancestral B cells.

(E) After scRNA-seq, the variable regions for the light and heavy immunoglobulin alleles are assembled, the isotypes are called, and the B cells are clustered into $k$ clonotypes. A multiple sequence alignment $\mathbf{A}_j$ is found for each clonotype $j$ and used to infer a set of input trees with maximum parsimony. The leaves of each input tree are labeled by isotypes $b$.

(F) TRIBAL jointly infers a B cell lineage tree $T_j^*$ for each clonotype $j$ and population-specific isotype transition probabilities $\mathbf{P}^*$ with maximum parsimony for MSA $\mathbf{A}_j$ and maximum likelihood for isotypes $b_j$.

See also STAR Methods.

use with maximum-likelihood inference via IgPhyML. While maximum-likelihood methods, such as IgPhyML, provide a single-point estimate of the lineage tree that maximizes the likelihood of the observed BCR sequences, they do not provide a complete picture of the uncertainty surrounding this estimate. Bootstrapping or the application of computationally intensive Bayesian methods is therefore needed to properly assess the extent of phylogenetic uncertainty.

Another important difference between B cell and species evolution is the lower mutation rate and the relatively short length of the BCR sequence ($\approx 600$ bp). These properties imply that maximum parsimony inference methods are viable[11,24,25] but typically result in a large solution space of many plausible phylogenies for the same data. In addition, these solutions exhibit additional tree uncertainty in the form of polytomies. Consequently, the inference of a B cell lineage tree from the BCR heavy- and light-chain sequences alone yields a high degree of phylogenetic uncertainty. Previously, sequence abundance has been utilized by both GCTree[11] and ClonalTree[12] on bulk RNA-seq data to resolve phylogenetic uncertainty. However, sampling limitations associated with scRNA-seq may dilute the

sequence abundance signal by yielding very few identical BCR sequences.

A critical advantage of scRNA-seq is that we can simultaneously measure both the BCR sequence and expressed antibody class of an individual B cell. The antibody class is a useful marker of a genetic process, known as class switch recombination (CSR), which seeks to diversify the role of B cells in the adaptive immune response by altering the antibody's functional class, or isotype, via genetic modification of the BCR isotype genes (Figures 1B and 1C).[4] When a B cell undergoes class switching from its current isotype to a new isotype, any heavy-chain constant-region locus between the current isotype and the new isotype in the genome is cut out or removed via a recombination process (Figure 1B). Consequently, CSR is an irreversible process, and the isotype of a B cell offers a distinct milestone in its evolutionary history. Therefore, the inclusion of isotype measurements from scRNA-seq as well as incorporating the known biological constraints of CSR into the problem of B cell lineage inference have the potential to help minimize phylogenetic uncertainty in terms of both reducing the size of the solution space and yielding more refined B cell lineage trees with fewer polytomies.

In this work, we present TRIBAL (tree inference of B cell clonal lineages) (Figure 1D). TRIBAL utilizes both the BCR sequence and isotype information from sequenced cells to infer a B cell lineage tree that jointly models the evolutionary and genetic processes of SHM and CSR. Additionally, TRIBAL infers the underlying isotype transition probabilities, providing valuable insight into the dynamics of CSR (Figure 1D). We demonstrate the accuracy of TRIBAL on simulated data and show that it is effective on experimental single-cell data generated from the 5′ 10× Genomics platform. TRIBAL is open source and has the potential to improve our understanding of vaccine responses, track disease progression, and identify therapeutic antibodies.

## DESIGN

To comprehensively model the evolutionary history of a collection of $n$ B cells clustered into $k$ clonotypes, TRIBAL aims to solve the following problem.

Problem 1 (B cell lineage forest inference [BLFI]): given multiple sequence alignments $\mathbf{A}_1, \ldots, \mathbf{A}_k$ and isotypes $\mathbf{b}_1, \ldots, \mathbf{b}_k$ for $k$ clonotypes, find isotype transition probabilities $\mathbf{P}^*$ for $r$ isotypes and lineage trees $T_1^*, \ldots, T_k^*$ for $(\mathbf{A}_1, \mathbf{b}_1), \ldots, (\mathbf{A}_k, \mathbf{b}_k)$ whose nodes are labeled by sequences $\alpha_1^*, \ldots, \alpha_k^*$ and isotypes $\beta_1^*, \ldots, \beta_k^*$, respectively, so that $\sum_{j=1}^{k} \text{SHM}(T_j^*, \alpha_j^*)$ is minimum, and then

$$\prod_{j=1}^{k} \text{CSR}(T_j^*, \beta_j^*, P^*)$$ is maximum.

The formal definition of a clonotype is a set of B cells that all descend from the same naive BCR, sharing identical genes in both the heavy- and light-chain variable regions. TRIBAL has two inputs. First, an MSA $\mathbf{A}_j$ is generated for each clonotype $j$ by concatenating the DNA sequences of the variable regions of the heavy and light chain of the BCR of the $n_j$ clonal B cells that descend from the same naive B cell post V(D)J recombination with sequence $a_{j,0}$. Second, isotypes $b_i \in [r] = \{1, \ldots, r\}$ are determined using tools such as Cell Ranger.[26] For humans, there are $r = 8$ isotypes linearly encoded from 1 to 8 as immunoglobulin M (IgM)/IgD, IgG3, IgG1, IgA1, IgG2, IgG4, IgE, and IgA2, whereas for mice there are $r = 7$ isotypes linearly encoded as IgM/D, IgG3, IgG1, IgG2b, IgG2c (2a), IgE, and IgGA.

TRIBAL infers a lineage tree $T_j$ for the $n_j$ B cells of each clonotype $j$, describing the joint evolution of the given DNA sequences $\mathbf{A}_j = [\mathbf{a}_{j,0}, \mathbf{a}_{j,1}, \ldots, \mathbf{a}_{j,n_j}]^\top$ and isotypes $b_j = [b_{j,0}, b_{j,1}, \ldots, b_{j,n_j}]^\top$. Specifically, $T_j$ is a rooted tree whose nodes $v$ are labeled by a DNA sequence $\alpha(v)$ and isotype $\beta(v)$ so that the root $v_0$ is labeled by $\alpha(v_0) = a_0$ and $\beta(v_0) = b_0 = 1$. On the other hand, the $n_j$ leaves $L(T_j) = \{v_1, \ldots, v_{n_j}\}$ are labeled by DNA sequence $\alpha(v_i) = a_{j,i}$ and isotype $\beta(v_i) = b_{j,i}$ for each B cell $i \in [n_j]$. In addition, due to the irreversibility of CSR, the isotype $\beta(u)$ of an ancestral cell $u$ must be less than or equal to the isotype $\beta(v)$ of its descendants $v$; i.e., $\beta(u) \leq \beta(v)$ for all edges $(u, v) \in E(T_j)$.

Lineage trees typically have shallow depth due to the limited number of mutations introduced during SHM,[11,24,25] making unweighted parsimony a reasonable evolutionary model for SHM. Thus, $\text{SHM}(T, \alpha)$ counts the total number of nucleotide substitutions in the lineage tree $T$ labeled by sequences $\alpha$. To model CSR, we use isotype transition probabilities $\mathbf{P} = [p_{s,t}]$ that cap-

ture the conditional probability of a descendant isotype $t$ given the isotype of its parent $s$ subject to irreversible isotype evolution; i.e., $p_{s,t} \geq 0$, $p_{s,t} = 0$ if $s > t$, and $\sum_{t=1}^{r} p_{s,t} = 1$ for all isotypes $s \in [r]$. Using independence along the edges $E(T)$ of a lineage tree $T$ allows us to define the joint likelihood $\text{CSR}(T, \beta, \mathbf{P})$ of the observed isotypes $b$ for isotype transition probabilities $\mathbf{P}$ and any lineage tree $T$ whose leaves have isotypes $b$ as $\prod_{(u,v) \in E(T)} p_{\beta(u), \beta(v)}$.

TRIBAL uses three key ideas to effectively solve the BLFI problem. First, a significant barrier to solving the BLFI problem is that isotype transition probabilities $\mathbf{P}$ are unknown and need to be inferred. While there have been experimental studies that estimate these quantities under specific biological conditions,[27] there currently exist no computational methods to directly infer these probabilities from a sequencing experiment. We reason that, under many experimental conditions, the transition probabilities will be shared across clonotypes, increasing our power to accurately estimate these parameters.

Second, the lexicographical ordering of the two objectives—optimizing for SHM followed by CSR—enables one to use the following two-stage approach (Figure 1C). In the first stage, we use existing maximum parsimony methods to generate a set $\mathcal{T}$ of input trees—also called a maximum parsimony forest—for each clonotype so that each tree $T \in \mathcal{T}$ minimizes the objective $\text{SHM}(T, \alpha)$. To do so, we provide these methods only the sequence information $\mathbf{A}$ to enumerate a solution space $\mathcal{T}$ of trees whose nodes are labeled by sequences $\alpha_1, \ldots, \alpha_{|\mathcal{T}|}$. In the second stage, we incorporate isotype information $b$ to further operate on the set $\mathcal{T}$ and additionally optimize $\text{CSR}(T, \beta, \mathbf{P})$ in a manner that maintains optimality of the SHM objective. We note that a lexicographically optimal lineage tree $T^*$ does not necessarily need to be an element of $\mathcal{T}$, but instead, it suffices that the evolutionary relationships in tree $T^*$ are a refinement of the evolutionary relationships described by some tree $T$ among the set $\mathcal{T}$ of input trees. More specifically, a refinement $T'$ of tree $T$ is obtained by zero or more EXPAND operations so that EXPAND $(v)$ results in splitting node $v$ into $v$ and $v'$, joining them with an edge $(v, v')$ and then reassigning a (potentially empty) subset of the children of $v$ to be children of $v'$. Importantly, one can obtain a refinement $T'$ of $T$ maintaining the SHM objective; i.e., $\text{SHM}(T, \alpha) = \text{SHM}(T', \alpha')$ by setting $\alpha(v') = \alpha(v)$ for each node $v'$ of $T'$ obtained via the EXPAND operation applied to node $v$ of $T$. Therefore, our sought lineage tree $T^*$ that first optimizes SHM and then CSR must be a refinement of some tree $T$ in the set $\mathcal{T}$ of unrefined trees with optimal SHM scores.

The third key idea is that the inference of optimal lineage trees $T_1^*, \ldots, T_k^*$ is conditionally independent when given isotype transition probabilities $\mathbf{P}$. This motivates the use of a coordinate ascent algorithm where we randomly initialize isotype transition probabilities $\mathbf{P}^{(1)}$ (Figure 2A). Then, at each iteration $\ell$, we use isotype transition probabilities $\mathbf{P}^{(\ell)}$ and the input set $\mathcal{T}_j$ of trees to independently infer an optimal lineage tree $T_j^{(\ell)}$ for each clonotype $j$ (Figure 2B). Briefly, this is achieved by solving the intermediate problem of finding the most parsimonious refinement of each tree $T$ in the maximum parsimony forest $\mathcal{T}$ utilizing a graph-based approach. This is then followed by estimating updated isotype transition probabilities $P^{(\ell+1)}$ given trees

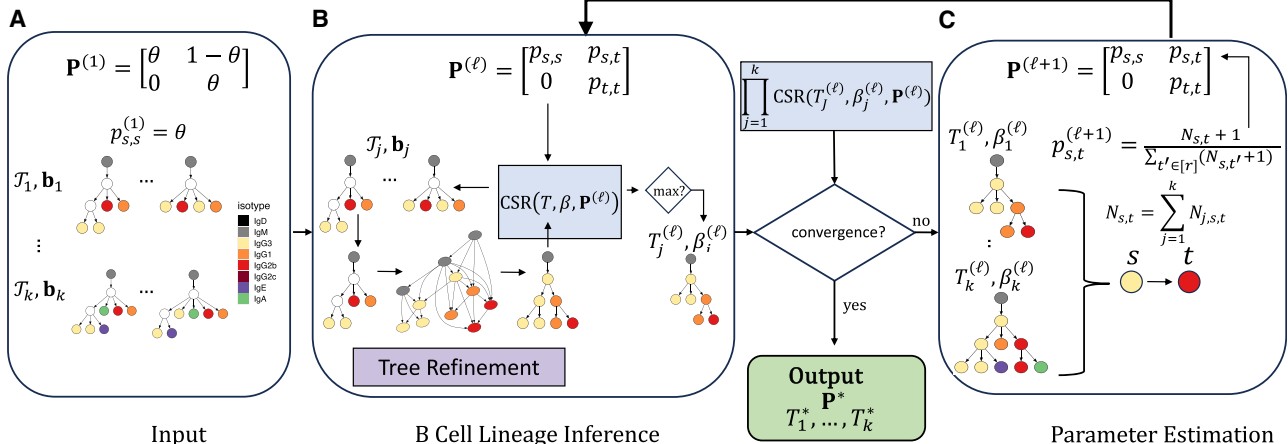

**A** | Input

**B** | B Cell Lineage Inference

**C** | Parameter Estimation

**Figure 2. TRIBAL infers B cell lineage forest $T_1^*, \ldots, T_k^*$ and isotype transitions $\mathbf{P}^*$ for $k$ clonotypes utilizing coordinate ascent**

(A) The inputs to TRIBAL are isotype transition probabilities $\mathbf{P}^{(1)}$, which are initialized given a parameter $\theta \in [0.5, 1]$ and a tuples $(\mathcal{T}_1, b_1) \ldots, (\mathcal{T}_k, b_k)$, where sets $\mathcal{T}_j$ are maximum parsimony trees for MSA $\mathbf{A}_j$ and $b_j$ are the observed isotypes of the $n_j$ cells of clonotype $j$.

(B) Conditioning on isotype transition probabilities $\mathbf{P}^{(\ell)}$, a B cell lineage tree $T_j^{(\ell)}$ with nodes labeled by isotypes $\beta_j^{(\ell)}$ is inferred for each clonotype $j$ by solving the MPTR problem for each tree in the input set $\mathcal{T}_j$.

(C) Convergence between $\prod_{j=1}^{k} \mathrm{CSR}(T_j, \beta_j, \mathbf{P})$ for iterations $\ell$ and $\ell$ is checked. If the difference has not converged, then isotype transition probabilities $\mathbf{P}^{(\ell+1)}$ are updated using maximum-likelihood estimation. If the difference has converged, then the current inferred B cell lineage forest and isotype transition probabilities $\mathbf{P}$ are output. Multiple restarts can be performed for varying values of $\theta$.

See also STAR Methods.

$T_1^{(\ell)}, \ldots, T_k^{(\ell)}$ via maximum-likelihood estimation (Figure 2C). We terminate upon convergence of our CSR objective or when exceeding a specified number of maximum iterations.

TRIBAL is implemented in Python 3, is open source (BSD-3-Clause license), and is available at https://github.com/elkebir-group/TRIBAL. See STAR Methods for a more detailed description of TRIBAL.

## RESULTS

### TRIBAL outperforms state-of-the-art methods across *in silico* experiments

We designed *in silico* experiments (STAR Methods) to evaluate TRIBAL with known ground-truth isotype transition probabilities **P** and lineage trees $T$ labeled by sequences $\alpha$ and isotypes $\beta$. Specifically, we extended an existing BCR phylogenetic simulator[24] that models SHM to additionally incorporate CSR. To that end, we generated isotype transition probabilities **P** with $r = 7$ isotypes (as in mice) under two different models of CSR. Both CSR models assume that the probability of not transitioning is higher than the probability of transitioning, but in the sequential model, there is clear preference for transitions to the next contiguous isotype, while in the direct model, the probabilities of contiguous and non-contiguous class are similar. Given **P**, we evolved isotype characters down each ground-truth lineage tree $T$.

We generated 5 replications of each CSR model for $k = 75$ clonotypes and $n \in \{35, 65\}$ cells per clonotype, resulting in 20 *in silico* experiments, yielding a total of 1,500 ground-truth lineage trees. In addition to comparing TRIBAL to existing methods, including neighbor joining (NJ),[28] ClonalTree,[12] dnapars,[20] dnaml,[20] and IgPhyML,[8] we also compared it to a version of

TRIBAL without tree refinement, denoted as TRIBAL-NO REFINEMENT (TRIBAL-NR). Although ClonalTree relies on genotype abundance, and our simulations include very few duplicated sequences, we included ClonalTree to benchmark TRIBAL against minimum spanning tree approaches, as also used by GlaMST.[29] Therefore, we ran ClonalTree both including and ignoring abundance data and selected the mode with the best performance, which was ignoring abundance data. To obtain the input set $\mathcal{T}_j$ of trees with maximum parsimony for each clonotype $j$, we utilized dnapars.[20] We refer the reader to STAR Methods for additional details on the simulations. In the following, we focus our discussion on *in silico* experiments with $j$ cells per clonotype (see Figure S1 for $n = 65$).

To evaluate the accuracy of isotype transition probability inference, we used Kullback-Leibler (KL) divergence[30] to compare the inferred transition probability distribution $\hat{\mathbf{p}}_s$ of each isotype $s$ to the simulated ground-truth distribution $p_s$—the lower the KL divergence, the more similar the two distributions. Since no existing methods infer isotype transition probabilities, we restricted this analysis to TRIBAL and TRIBAL-NR. Overall, we observed good concordance between simulated and TRIBAL-inferred isotype transition probabilities (Figure 3A). Specifically, TRIBAL had lower median KL divergence than TRIBAL-NR for all isotype starting states, except IgA, which is trivially 0, under both direct and sequential CSR models (direct: median of 0.15 vs. 0.73; sequential: median of 0.099 vs. 0.55). We observed improved performance of TRIBAL (but not for TRIBAL-NR) for $n = 65$ cells per clonotype (Figures S1 and S2).

To assess the sensitivity of TRIBAL to infer isotype transition probabilities with fewer than $k = 75$ clonotypes, we downsampled the 75 clonotypes to 25 and 50 clonotypes per

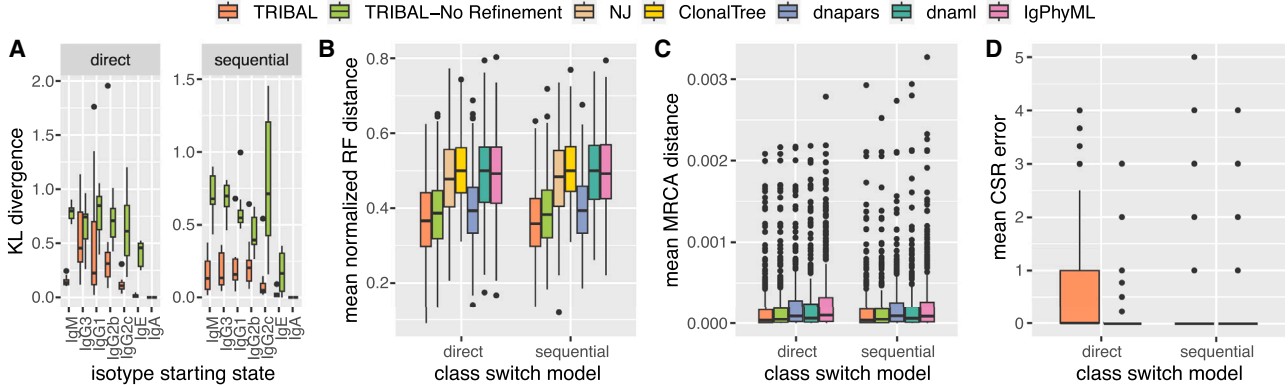

**Figure 3. TRIBAL accurately infers isotype transition probabilities on simulated data while outperforming existing methods on lineage tree inference**

Simulation results shown are for 5 replications with $k = 75$ clonotypes per replication and $n = 35$ cells per clonotype.

(A) KL divergence between inferred isotype transition probabilities and the reference ground-truth distribution.

(B) Mean RF distance between ground-truth and inferred lineages tree per clonotype.

(C) Mean MRCA distance (23) between ground truth and inferred lineage trees per clonotype. Note that whisker length is set to five times the interquartile range.

(D) Mean CSR error between ground truth and inferred B cells.

See also Figures S1–S3.

experiment. We observed similar trends for experiments with $k \in \{25, 50\}$ clonotypes, with TRIBAL continuing to outperform TRIBAL-NR while still achieving small KL divergences even as $k$ decreases (Figure S3). These findings demonstrate that tree refinement is key to accurately estimate isotype transition probabilities.

Next, we assessed the accuracy of lineage tree inference using the Robinson-Foulds (RF) distance[31] normalized by the total number of bipartitions in the ground truth $T$ and inferred lineage tree $\hat{T}$ (22). Since TRIBAL, TRIBAL-NR, and dnapars return multiple optimal solutions, we report the mean of the lineage tree inference metrics over all optimal solutions. To compute the RF distance for ClonalTree, which returns a minimum spanning tree (MST) of the sequenced B cells, we converted the MST to a lineage tree by adding a leaf node for each sequenced B cell representing an internal node (i.e., an extant ancestor of other B cells), and introducing an edge between the internal node and the newly added leaf node. We found that TRIBAL had the lowest mean normalized RF distance for both direct and sequential CSR models (Figure 3B). Overall, NJ (median: 0.48), ClonalTree (median: 0.5), dnaml (median: 0.5), and IgPhyML (median: 0.49) had the worst performance on normalized RF. Interestingly, even though the starting trees of dnapars are used by TRIBAL, both TRIBAL (median: 0.36) and TRIBAL-NR (median: 0.38) outperformed dnapars (median: 0.39), showing the importance of using isotype information to resolve phylogenetic uncertainty.

While normalized RF distance only assesses the accuracy of the tree topology, it is important to also assess the accuracy of the ancestral sequence reconstruction. To that end, we used a metric called Most Recent Common Ancestor (MRCA) distance (23) introduced by Davidsen and Matsen.[24] For any two simulated B cells (leaves), the MRCA distance is the Hamming distance between the MRCA sequences of these two B cells in both the ground-truth and inferred lineage trees. This distance

is then averaged over all pairs of simulated B cells (see STAR Methods and Figure S12A for additional details). We excluded NJ and ClonalTree from this analysis, as these distance-based methods do not infer ancestral sequences. Again, we report the mean of overall optimal solutions for TRIBAL, TRIBAL-NR, and dnapars. We found that TRIBAL outperformed all other methods (Figure 3C), achieving the lowest overall median MRCA distance ($3.46 \times 10^{-5}$), followed by TRIBAL-NR ($3.46 \times 10^{-5}$). IgPhyML had the worst performance with a median of $8.78 \times 10^{-5}$. Performance trends were consistent between methods across both CSR models.

Last, we assessed the accuracy of isotype inference. Since NJ, dnaml, dnapars, and IgPhyML do not infer isotypes, we excluded these methods from this analysis. ClonalTree, on the other hand, infers an MST so that sequenced B cells are inferred to be ancestral to other sequenced B cells. This permitted assessment of inferred class switching in these ClonalTree-inferred MSTs. We calculated the percentage of inferred trees by ClonalTree that induced violations of the class switching constraints; i.e., implying reversible evolution. We observed that 92% of the inferred ClonalTree MSTs contained at least one invalid isotype transition. Moreover, a mean of 21% of the edges in each ClonalTree MST were indicative of invalid isotype transitions. These results highlight the critical importance of incorporating isotype data when inferring B cell lineage trees in order to accurately infer the evolutionary relationships of a B cell clonal lineage. To assess the accuracy of the TRIBAL-inferred isotypes, we developed a new metric called CSR error, which is computed for each B cell $i$ and clonotype $j$ and is the absolute difference between the number of ground-truth class switches and inferred number of class switches that occurred along the evolutionary path from the root to the sequenced B cell (see STAR Methods and Figure S12B for additional details). We account for the presence of multiple solutions by taking the mean across solutions.

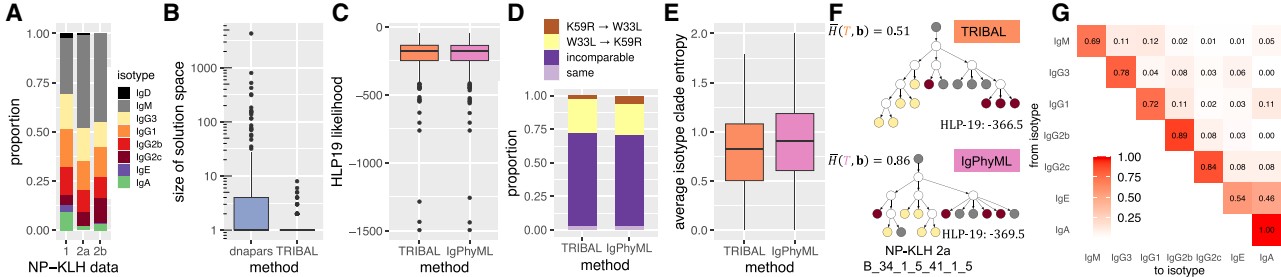

**Figure 4. Comparison between TRIBAL and IgPhyML on the NP-KLH data**
(A) The distribution of isotypes $b$ in each dataset.
(B) A comparison of the solution space of dnapars versus TRIBAL.
(C) The distribution of the HLP19 codon substitution likelihood[23] for lineage trees inferred by TRIBAL and IgPhyML.
(D) Observed distribution of evolutionary relationships between W33L and K59 in clonotypes where both mutations are present.
(E) The distribution of the average clade entropy with respect to an isotype labeling $b$ of the leaf set.
(F) A comparison of lineage trees inferred for clonotype NP-KLH-2a B_34_1_5_41_1_5, with the average isotype clade entropy $\overline{H}(T, b)$ reported for each inferred tree.
(G) TRIBAL-inferred isotype transition probabilities $\mathbf{P}$ for NP-KLH-1.
See also Figures S4–S6 and Table S1.

Both methods had a median CSR error of 0 for both the direct and sequential models (Figure 3D). Therefore, we utilized the third quartile for a more robust comparison. We found that, under the direct model, TRIBAL-NR (third quartile: 0) was the best-performing method, and TRIBAL had a third quartile of 1.

We observed a slight tendency of TRIBAL to overestimate the number of transitions due to the tree refinement step, while other methods tended to underestimate the number of transitions. This slight overestimation is likely due to utilizing the maximum-likelihood estimates of the inferred ancestral isotypes as opposed to considering the marginal distribution of ancestral isotype states for each node. In other words, for any given clonotype, it is difficult to infer whether the unobserved ancestral isotypes underwent direct or sequential class switching, but given multiple clonotypes, TRIBAL is able to more accurately tease out these relative frequencies in class-switching than TRIBAL-NR (Figure 3A). However, under a sequential model, refinement is particularly helpful in accurately capturing sequential state transitions; we found that TRIBAL was tied with TRIBAL-NR for the best performance (third quartile: 0). All other methods had similar performance between both CSR models for this metric.

### TRIBAL recapitulates known biology trends on an NP-KLH model affinity maturation system
We applied TRIBAL as well as IgPhyML to 10× Genomics 5′ scRNA-seq data of B cells extracted from mice immunized with nucleoprotein keyhole limpet hemocyanin (NP-KLH), a commonly used antigen in the study of antibody affinity maturation.[32] Our goal was to determine whether these methods recapitulate known patterns of B cell lineage evolution for this well-studied antigen using data from two studies and to compare the lineage trees inferred by each method. The first dataset (NP-KLH-1) was generated from C57BL/6 mice that were immunized with NP-KLH, and total germinal center B cells were extracted 14 days after immunization.[33] The other two datasets came from a single study in which C57BL/6 mice were immunized with NP-KLH (NP-KLH-2a and NP-KLH-2b), and NP-specific germinal center B cells were extracted 13 days after immunization.[34] We utilized the standard 10× Genomics Cell Ranger[26] single-cell bioinformatics pipeline to generate sequence $a_i$ and isotype $b_i$ for each cell $i$. We used Dandelion[35] to remove doublets, reassign alleles, and cluster the cells into clonotypes. Dandelion specifies clonotypes using the following ordered criteria for both heavy- and light-chain contigs as follows: (1) identical V and J gene usage, (2) identical junctional CDR3 amino acid length, and (3) CDR3 sequence similarity with the default setting for BCRs set to 85% amino acid sequence similarity based on Hamming distance. Network analysis is then used to assign clusters.[35] We identified clonotype MSAs $\mathbf{A}_1, \ldots, \mathbf{A}_k$ based on shared V(D)J alleles for the heavy chain using the Dowser package.[25] Finally, we excluded clonotypes with fewer than 5 cells. This yielded a total of $n = 2670$ sequenced B cells clustered into $k = 295$ clonotypes. We exclude methods that rely on sequence abundance as a key signal, such as GCTree[11] and ClonalTree,[12] as we observed very few duplicated sequences within each clonotype. Figure 4A shows the distribution of isotypes by dataset, and Table S1 includes a more detailed summary of each dataset.

We used dnapars[20] to infer TRIBAL's input set $\mathcal{T}_j$ for each clonotype $j$. We found that TRIBAL's use of isotype information significantly reduced the number of optimal solutions identified by dnapars (mean: 31.5 vs. 1.3; max: 4310 vs. 8) (Figure 4B). While IgPhyML, a maximum-likelihood method using the HLP19 codon substitution model,[8,23] infers only a single tree per clonotype, it is important to note that there might be multiple trees with maximum likelihood in the solution space. Indeed, we found high concordance of HLP19 likelihoods between the TRIBAL and IgPhyML inferred lineage trees, with a small overall mean absolute deviation of 0.97 (Figures 4C and S4). We even observed that TRIBAL had a greater likelihood than IgPhyML in 59.3% of the clonotypes. Thus, TRIBAL resulted in a significant reduction in the size of the solution space compared to the maximum parsimony method dnapars with similar (and sometimes better) HLP19 likelihood as IgPhyML, illustrating how

isotype information can be used to effectively reduce phylogenetic uncertainty.

Specifically, we categorized the relationship as K59R → W33L if K59R was ancestral to W33L, W33L → K59R if W33L was ancestral to K59R, as incomparable if W33L and K59R occurred on distinct lineages of the tree, and as same if they were introduced on the same edge of the lineage tree. Indeed, we confirmed the tendency for mutual exclusivity of W33L and K59R by finding that the proportion of pairwise introductions categorized as incomparable was 0.69 and 0.67 for TRIBAL and IgPhyML, respectively (Figure 4D). Additionally, it has been suggested that W33L mutations appear relatively early during the anti-NP response, whereas the K59R and S66N mutations typically appear later in the evolutionary history.[36] Defining level as the length of the shortest path from the MRCA of all B cells, we observed that W33L occurred at a median level of 1 for both TRIBAL and IgPhyML, while the K59R and S66N mutations occurred at a median level of 2 for both methods (Figure S5). This indicated that W33L was typically introduced earlier in the evolutionary history of a clonotype than K59R and S66N. Thus, both TRIBAL and IgPhyML trees recapitulate expected mutation patterns for this model system.

We next assessed the extent of agreement with isotype information. While TRIBAL infers isotype labels of ancestral nodes, IgPhyML does not have this capability. Therefore, we developed a new metric called average isotype clade entropy, which is computed with respect to the isotype labeling of the leaf set. For this metric, we compute the entropy of clade $u$ in tree $T$ with respect to all isotype leaf labels that are descendants of node $u$, taking the average entropy over all non-trivial clades, which excludes the root and the leaves (STAR Methods). As IgPhyML returns bifurcating trees, we collapse edges with zero branch length for a fairer comparison of this metric. We observed lower average isotype clade entropy for the TRIBAL (median: 0.82) versus IgPhyML (median: 0.91) inferred trees (Figure 4E). Figure 4F depicts the lineage tree inferred by TRIBAL and IgPhyML for the NP-KLH-2a dataset (clonotype B_34_1_5_41_1_5). The TRIBAL-inferred tree for this clonotype had lower isotype clade entropy than IgPhyML (TRIBAL: 0.51 vs. IgPhyML: 0.86) while also resulting in a greater HLP19 likelihood (TRIBAL: −366.5 vs. IgPhyML: −369.5). Thus, we find that the trees identified by TRIBAL are in better agreement with the leaf isotypes than IgPhyML.

In addition to the inferred B cell lineage trees, TRIBAL also inferred isotype transition probabilities **P** for each dataset (Figures 4G and S6). All three inferred isotype transition probability matrices more closely matched a CSR model of direct switching as opposed to a strictly sequential model. To compare the consistency of these estimates across datasets, we computed the Jensen-Shannon divergence (JSD) between the distribution of isotype transition probabilities for each isotype starting state IgM through IgG2c for each dataset pair. We observed low JSD (median: 0.029) across a total of 15 pairwise comparisons, suggesting consistent estimates between isotype transition probabilities.

In summary, these analyses show that the inclusion of isotype information and tree refinement has the potential to yield high-quality lineage tree inference, even under a simpler model of SHM; i.e., parsimony. Moreover, the TRIBAL-inferred lineage trees additionally optimize for CSR, yielding lower isotype entropy partitions of the leaf set than IgPhyML. Finally, the additional inference of isotype transition probabilities **P** has the potential to distinguish between direct versus sequential switching events.

## TRIBAL infers B cell lineage trees with more parsimonious class switching on an age-associated B cell dataset

Next, we evaluated TRIBAL on three scRNA-seq datasets with V region sequencing that investigated the relationship between age-associated B cells (ABCs) and autoimmune disorders.[37] For each dataset, B cells were extracted from the spleen of a female MRL/lpr mouse and sequenced using 10×5′ scRNA-seq. The data were processed by the 10× Cell Ranger[26] single-cell bioinformatics pipeline to generate sequence $a_i$ and isotype $b_i$ for each cell $i$.

Nickerson et al.[37] identified clonotype MSAs $A_1, …, A_k$ based on shared V(D)J alleles for the heavy chain using the Dowser package[25] and inferred B cell lineage trees using IgPhyML for each clonotype. After filtering out clonotypes with fewer than 5 sequences, we retained 599 B cells and 54 clonotypes across the three datasets (Table S2). Figure 5A shows the proportion of isotypes and annotations by mouse for the retained B cells. Of these 54 clonotypes, 35 had more than one distinct isotype across the sequenced B cells, with a median of 3 distinct isotypes per clonotype.

We ran TRIBAL separately on each of the three mouse datasets, obtaining a maximum parsimony forest $\mathcal{T}_j$ for each clonotype $j$ via dnapars. Similar to our NP-KLH analysis, we found that TRIBAL effectively utilized the additional isotype data to reduce the number of optimal solutions identified by dnapars (mean: 8.1 vs. 1.3, max: 165 vs. 4) (Figure 5B). The HLP19 likelihood of the TRIBAL inferred lineage trees had high concordance with the IgPhyML inferred trees (mean absolute deviation: 0.97), with TRIBAL yielding a higher likelihood for 53% of the clonotypes (Figures 5C and S7). The average isotype clade entropy for the 35 clonotypes with more than one distinct isotype was significantly lower for TRIBAL than for IgPhyML (median: 0.49 vs. 0.77) (Figure 5D). An example comparison is shown in Figures 5E and 5F for clonotype Mouse-1 775. The tree refinement step of TRIBAL yielded a tree with a significantly lower average isotype clade entropy when compared to IgPhyML (0.65 vs. 1.2), while both trees had identical HLP19 likelihoods (−41.4). Finally, we observed that the isotype transition probabilities reveal evidence of both direct and sequential switching of isotypes (Figure S8).

In summary, both TRIBAL and IgPhyML yield lineage trees with very similar HLP19 likelihoods, giving support to the validity of the TRIBAL-inferred lineage trees in terms of sequence evolution. However, TRIBAL jointly optimizes evolutionary models for both SHM and CSR, yielding trees with lower average isotype clade entropy.

## TRIBAL infers B cell lineage trees for SARS-CoV-2 mRNA-1273 vaccine single-cell data

Finally, to further demonstrate the capabilities of TRIBAL on human data, we ran TRIBAL on a longitudinal single-cell analysis of

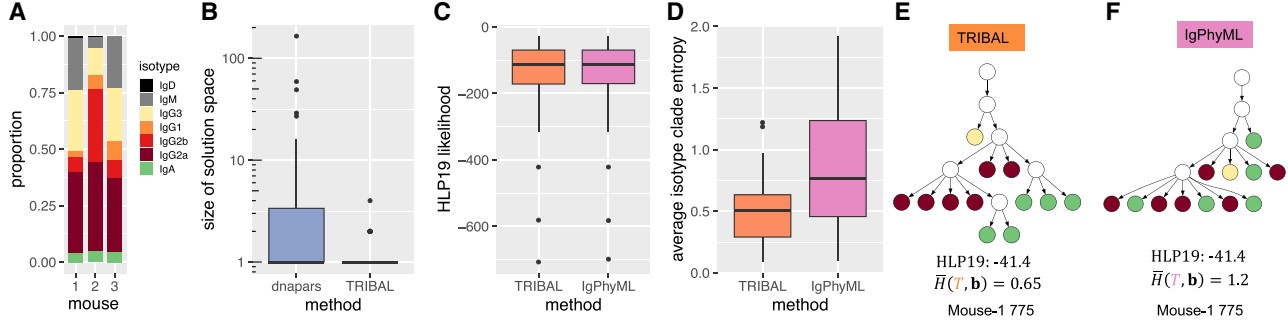

**Figure 5. Comparison between TRIBAL and IgPhyML on ABC data**
(A) Distribution of B cell isotypes.
(B) A comparison of the solution space of dnapars versus TRIBAL.
(C) The distribution of the HLP19 codon substitution likelihood[23] for lineage trees inferred by TRIBAL and IgPhyML.
(D) Comparison of average isotype clade entropy for TRIBAL versus IgPhyML.
(E and F) Comparison of inferred B cell lineage trees by TRIBAL (E) and IgPhyML (F) for clonotype Mouse-1 775; see (A) for a color legend.
See also Figures S7 and S8 and Table S2.

immune response to the severe acute respiratory syndrome coronavirus 2 (SARS-CoV-2) mRNA-1273 vaccine in infection-naive individuals.[38] BCR-annotated contigs and clonotype consensus sequences for each time point were obtained from the original study, and all time points were pooled together. We retained only cells that contained one productive heavy chain and one productive light chain. To map identical clonotypes across different time points, new clonotype IDs were assigned to the pooled data using the original clonotype consensus sequences. Full-length variable sequences for each BCR were then constructed by concatenating various framework and complementary-determining regions. Consensus sequences of heavy- and light-chain variable sequences across a clonotype were used as root sequences. Following this, the data contained 138,307 B cells from 131,460 clonotypes. We then further filtered the data to include clonotypes with at least 5 B cells, resulting in 2,508 B cells in 207 distinct clonotypes.

Of the $r = 8$ human isotypes ordered as IgM/D, IgG3, IgG1, IgA1, IgG2, IgG4, IgE, and IgA2, the data contained 7 distinct isotypes, with no B cells having the isotype IgE. The distribution of isotypes is shown in Figure 6A, with IgG1 (0.35) and IgA1 (0.33) having the largest proportions. Of these 207 clonotypes 61.3% (127) contained one unique isotype, 25.6% (53) contained two unique isotypes, and 13.0% (27) had at least three unique isotypes. We aligned the sequences of each clonotype to the inferred germline sequences using MAFFT v.7.5[39] and then ran dnapars[20] to obtain a maximum parsimony forest. We benchmarked TRIBAL against TRIBAL-NR in order to highlight the importance of resolving phylogenetic uncertainty.

First, we compared the size of the solution space of both TRIBAL and TRIBAL-NR versus the maximum parsimony forests obtain via dnapars. We only included clonotypes that contained at least two distinct isotypes (80 clonotypes). While all three methods had a median of 1 solution, TRIBAL had the smallest mean (4.55) compared to TRIBAL-NR (5.04) and dnapars (23.8) (Figure 6B). This demonstrates that the inclusion of isotype data is useful to resolve phylogenetic uncertainty by reducing the average size of the solution space.

Although it is well established that IgG and IgA antibody levels are enriched following SARS-CoV-2 infection or vaccination,[40–44] TRIBAL is capable of offering further insight into the dynamics of class switching. In particular, the inferred isotype transition probabilities (Figure 6B) indicate a high probability of direct class switching from IgM to both IgG1 (0.113) and IgA1 (0.061). In addition, we see evidence of sequential switching from IgG1 to IgA1 (0.073) as well as direct switching from IgG1 to IgG2 (0.031) and IgG2 to IgA2 (0.079). sciCSR,[45] a method that uses germline "sterile" transcripts to infer CSR dynamics, also observed these direct and sequential class switch patterns in a different SARS-CoV-2 vaccine single-cell dataset.[46]

As both TRIBAL and TRIBAL-NR make use of isotype data, we next assessed whether polytomy refinement yielded more cohesive partitions of the leaves with respect to isotype. We compared the average isotype clade entropy (STAR Methods) of the 9 clonotypes that had more than 2 distinct isotypes and more than one maximum parsimony tree. In the case of multiple optimal solutions, we took the mean of the average isotype clade entropy for all trees in the solution space (Figure 6D). We found that TRIBAL yielded a lower average isotype clade entropy (median: 0.69) compared to TRIBAL-NR (median: 0.87). A lower average isotype clade entropy implies a more plausible evolutionary history with respect to class switching because it correlates with fewer independent class switch events. Figure 6D, which compares the inferred B cell lineage trees of TRIBAL and TRIBAL-NR, highlights the utility of tree refinement.

Both methods inferred a B cell with isotype IgG1 as the MRCA of all sequenced B cells. However, the MRCA of the TRIBAL-NR has an outdegree of 12, implying that 10 independent class switch events occurred. In particular, it indicates that three independent class switch events occurred from IgG1 to IgA2. However, the isotype transition probabilities, which were inferred using all isotypes (Figure 6C), indicate that direct switching from IgG1 to IgA2 is a low-probability event (0.004). In contrast, the TRIBAL-inferred lineage tree indicates only two independent class switch events: one to IgA1 (0.073) and one to IgG2

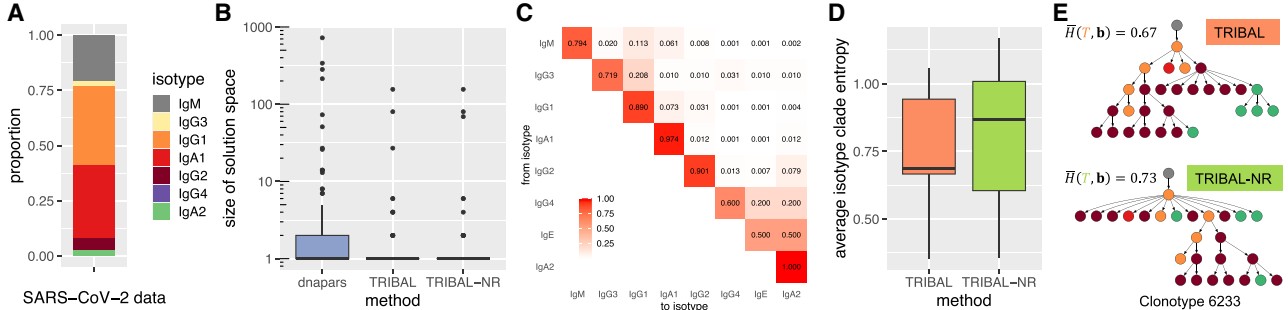

**Figure 6. Comparison between TRIBAL and TRIBAL-NR on longitudinal SARS-CoV-2 vaccine response data**
(A) Distribution of B cell isotypes.
(B) Comparison of the solution space of dnapars versus TRIBAL and TRIBAL-NR.
(C) Inferred TRIBAL isotype transition probabilities.
(D) Comparison of average isotype clade entropy for TRIBAL and TRIBAL-NR.
(E) Comparison of inferred B cell lineage trees by TRIBAL and TRIBAL-NR for clonotype 6233.

(0.031). Thus, the TRIBAL lineage tree is more consistent with the class switch dynamics inferred across all clonotypes.

In summary, the inclusion of isotype data into the B cell lineage inference problem is valuable for reducing the size of the solution space and enhancing our understanding of CSR following vaccination. However, simply incorporating isotype data alone is insufficient to yield B cell lineage trees that plausibly model CSR dynamics. These analyses highlight the critical importance of utilizing both isotype and tree refinement to reduce phylogenetic uncertainty and obtain lineage trees that accurately reflect CSR dynamics during the adaptive immune response to SARS-CoV-2 immunization.

## DISCUSSION

The development and application of methods for inferring B cell lineage trees and isotype transition probabilities from scRNA-seq data are crucial for improving our understanding of the immune system and adaptive immune responses, such as vaccine responses. In this work, we introduced TRIBAL, a method to infer B cell lineage trees and isotype transition probabilities from scRNA-seq data.

TRIBAL makes use of existing maximum parsimony methods to optimize an evolutionary model for SHM, then incorporates isotype data to find the most parsimonious refinement (i.e., maximizing the CSR likelihood), among the input set of trees. The main innovation of TRIBAL is that the inclusion of isotype data allows us to reduce phylogenetic uncertainty with respect to both the number of optimal solutions and refinement of the evolutionary relationships between B cells. Furthermore, TRIBAL provides isotype transition probabilities and inferred ancestral isotypes, enabling researchers to study CSR dynamics from a single time point and model the interplay between SHM and CSR during the adaptive immune response.

We demonstrated the effectiveness of TRIBAL via *in silico* experiments and on experimental data. On *in silico* experiments, we highlighted the importance of tree refinement for both accurately estimating isotype transition probabilities and lineage tree inference. Furthermore, we demonstrated on experimental data

that TRIBAL returns lineage trees that have similar HLP19 likelihoods despite utilizing a less complex model for sequence evolution but yield a reduction in the entropy of the isotype leaf labeling. Our integration of additional information suggests that TRIBAL could also be used with other types of information, such as CRISPR-Cas9 barcode editing, to better elucidate developmental lineages.[47]

There are several directions for future research that we anticipate. First, integration of germline "sterile" transcripts may offer a way to initialize the TRIBAL inferred isotype transition probabilities.[45] Second, many existing B cell lineage inference methods, such as IgPhyML, yield multifurcating trees when zero length branches are collapsed. There exists an opportunity to combine likelihood- or distance-based inference methods with the tree refinement step of TRIBAL. Third, the most parsimonious tree refinement (MPTR) problem has a more general formulation with the potential for wider applications beyond the problem of B cell lineage inference. For example, sample location is useful in refining tumor phylogeny with polytomies.[13] On a related note, we hypothesize that there are special cases of the MPTR problem and its more general formulation that are in P. Such special cases may include a weight matrix with unit costs and an upper triangular weight matrix that adheres to the triangle inequality.

Fourth, the assumption that a single isotype transition probability matrix is shared by all clonotypes could be relaxed to allow the inference of multiple matrices per experiment and an assignment of clonotypes to an inferred matrix. Fifth, TRIBAL could also be extended to jointly model SHM, CSR, and B cell states (e.g., naive, memory) derived from the sequenced transcriptome to provide a more comprehensive reconstruction of B cell evolution during the adaptive immune response. Sixth, more robust evolutionary models for SHM could be used to capture the presence of complex mutations, such as insertions or deletions, introduced during affinity maturation.[48,49]

Finally, future versions of TRIBAL could attempt to identify and mitigate sequencing and preprocessing errors; for example, by allowing inaccurately clonotyped B cells to move between B cell lineage trees.

Recent advancements in single-cell sequencing technologies have significantly enhanced the efficiency of cell capture and high-throughput profiling. These improvements now allow for the capture and sequencing of up to one million B cells from multiple patient cohorts (https://www.parsebiosciences.com/datasets/bcr-sequencing-of-1-million-healthy-and-diseased-samples-in-a-single-experiment/), paving the way for deeper insights into cellular diversity and disease mechanisms. This increase in cell numbers presents new computing challenges. However, as TRIBAL is the only method to model both SHM and CSR, it is well suited to help researchers understand the relationship between SHM and CSR and to elucidate CSR dynamics within and across different disease cohorts at large scale.

### Limitations of the study

There are a number of limitations of this study. First, our method assumes that all clonotypes share the same isotype transition probabilities. Whether such an assumption holds in practice will be dependent on the experimental design. We expect such an assumption to hold for samples collected at the same approximate location. Second, several upstream steps directly influence our ability to better reconstruct B cell lineage trees, including preprocessing of scRNA-seq data using tools such as Cell Ranger,[26] Dandelion,[50] and Dowser.[25] While these methods have been optimized to minimize the impact of sequencing errors and noise, experimental design choices such as sequencing depth or sample diversity may impact the output of these methods and, subsequently, the input data to TRIBAL. We recommend adherence to best practices for both the design of the scRNA-seq experiment and of the preprocessing methods utilized to improve accuracy of the TRIBAL input data. Finally, the accuracy of the inference of isotype transition probabilities will improve as the number of clonotypes increases. Additional experiments are required to identify a lower bound on the number of clonotypes needed for reliable isotype transition probability estimation.

### STAR★METHODS

Detailed methods are provided in the online version of this paper and include the following:

- KEY RESOURCES TABLE
- RESOURCE AVAILABILITY
  - Lead contact
  - Materials availability
  - Data and code availability
- METHOD DETAILS
  - TRIBAL output and optimization criteria
  - The TRIBAL algorithm
  - Initialization of isotype transition probabilities
  - Optimizing B cell lineage trees given isotype transition probabilities
  - Optimizing isotype transition probabilities given B cell lineage trees
  - B cell lineage forest inference
  - Combinatorial characterization of the most parsimonious tree refinement problem
  - Simulation details
  - SHM simulation and benchmarking
  - CSR simulation
  - Inference using TRIBAL
  - *In silico* study performance metrics
  - Average clade entropy for a leaf labeling

### SUPPLEMENTAL INFORMATION

### ACKNOWLEDGMENTS

We thank Harinder Singh, Ken Hoehn, Mark Shlomchik, and Margie Ackerman for insightful discussions. This work was partially supported by the NIH grant DP2AI177884 (to A.A.K.) and by the National Science Foundation grant CCF-2046488 (to M.E.-K.). This work used resources, services, and support provided via the Greg Gulick Honorary Research Award Opportunity supported by a gift from Amazon Web Services. This work started at the Computational Genomics Summer Institute (CGSI) 2021.

### AUTHOR CONTRIBUTIONS

Conceptualization, M.E.-K. and A.A.K; methodology, L.L.W., M.E.-K., and A.A.K.; investigation, L.L.W., D.R., M.S.R., Y.Q., M.E.-K., and A.A.K.; software, L.L.W.; data curation, D.R.; writing – original draft, L.L.W., M.E-.K., and A.A.K.; writing – review & editing, L.L.W., D.R., M.E.-K., and A.A.K.; funding acquisition, M.E.-K. and A.A.K.; supervision, M.E.-K. and A.A.K.

### DECLARATION OF INTERESTS

The authors declare no competing interests.

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

## Technology

## STAR★METHODS

### KEY RESOURCES TABLE

| REAGENT or RESOURCE | SOURCE | IDENTIFIER |
|---|---|---|
| **Deposited data** | | |
| NP-KLH dataset | Reiman et al.[33] | GEO: GSE171867 |
| ABC dataset | Nickerson et al.[37] | GEO: GSE202110 |
| SARS-CoV-2 dataset | de Assis et al.[38] | GEO: GSE219098 |
| **Software and algorithms** | | |
| Python(v3.9.15) | Python Software Foundation | https://www.python.org/ |
| R(v4.3.1) | R Core Team | https://www.r-project.org/ |
| Docker(v1.10.2) | Docker | https://www.docker.com |
| MAFFT(v7.5) | Katoh et al.[39] | https://mafft.cbrc.jp/alignment/software/ |
| PHYLIP(v3.6) | Felsenstein[20] | https://phylipweb.github.io/phylip/ |
| Gurobi(v10.0) | Gurobi Optimization | https://www.gurobi.com |
| dowser | Hoehn et al.[25] | https://dowser.readthedocs.io/en/latest/ |
| IgPhyML | Hoehn et al.[8,23] | https://igphyml.readthedocs.io/en/latest/ |
| Dandelion | Stephenson et al.[50] | https://sc-dandelion.readthedocs.io/en/latest/index.html |
| ClonalTree | Abdollahi et al.[12] | https://github.com/julibinho/ClonalTree |
| TRIBAL | This paper | https://doi.org/10.5281/zenodo.12741290 |

### RESOURCE AVAILABILITY

#### Lead contact
Further information and requests for resources and reagents should be directed to and will be fulfilled by the lead contact, Mohammed El-Kebir (melkebir@illinois.edu).

#### Materials availability
This study did not generate new unique reagents.

#### Data and code availability
- This paper analyzes existing, publicly available data. These accession numbers for the datasets are listed in the key resources table. Simulated and processed data are available at https://doi.org/10.5281/zenodo.12746319.
- All original code has been deposited at https://elkebir-group.github.io/TRIBAL/. The repository is archived on Zenodo (https://doi.org/10.5281/zenodo.12741290).
- Any additional information required to reanalyze the data reported in this paper is available from the corresponding authors upon request.

### METHOD DETAILS

#### TRIBAL output and optimization criteria
To comprehensively model the evolutionary history of $n$ B cells, we aim to construct a B cell lineage tree $T$ that jointly describes the evolution of the B cells' DNA sequences $\mathbf{A} = [a_0, a_1, \ldots, a_n]^\top$ under somatic hypermutation and affinity maturation and their isotypes $b = [b_0, b_1, \ldots, b_n]^\top$ via class switch recombination. As such, each node $v$ of $T$ will be labeled by a sequence $\alpha(v) \in \Sigma^m$ and isotype $\beta(v) \in [r]$. In particular, the root $v_0$ will be labeled by $\alpha(v_0) = a_0$ and $\beta(v_0) = b_0 = 1$ while the $n$ leaves $L(T) = \{v_1, \ldots, v_n\}$ of $T$ will be labeled by sequence $\alpha(v_i) = a_i$ and isotype $\beta(b_i) = b_i$ for each $i \in [n]$. A key property of isotype switching is that it is *irreversible*. As such, the isotype $\beta(u)$ of an ancestral cell $u$ must be less than or equal to the isotype $\beta(v)$ of its descendants $v$. More formally, we have the following definition of a B cell lineage tree.

**Definition 1.** A rooted tree $T$ whose nodes are labeled by sequences $\alpha : V(T) \to \Sigma^m$ and isotypes $\beta : V(T) \to [r]$ is a *B cell lineage tree* for MSA $\mathbf{A} = [a_0, a_1, \ldots, a_n]^\top$ and isotypes $b = [b_0, b_1, \ldots, b_n]^\top$ provided (i) $T$ has $n$ leaves $L(T) = \{v_1, \ldots, v_n\}$ such that each leaf $v_i \in L(T)$ is labeled by sequence $\alpha(v_i) = a_i$ and isotype $\beta(v_i) = b_i$, (ii) the root node $v_0$ of $T$ is labeled by sequence $\alpha(v_0) = a_0$ and isotype $\beta(v_0) = b_0$, and (iii) for all nodes $u, v \in V(T)$ such that is ancestral to $v$ it holds that $\beta(u) \leq \beta(v)$.

In the following, we will refer to B cell lineage trees as *lineage trees*. Lineage trees typically have shallow depth due to the limited number of mutations introduced during SHM, making parsimony a reasonable evolutionary model for SHM.[11,24,25] Given a lineage tree $T$, the SHM parsimony score is computed as,

$$\mathrm{SHM}(T, \alpha) \;=\; \sum_{(u,v) \in E(T)} D(\alpha(u), \alpha(v)), \tag{Equation 1}$$

where $D(\alpha(u), \alpha(v))$ is the Hamming distance[51] between sequences $\alpha(u)$ and $\alpha(v)$. However, one common challenge of using parsimony to model SHM is that it often results in a large number of candidate lineage trees with equal optimal parsimony score. In addition, many inferred lineage trees contain *polytomies*, or internal nodes with out-degree greater than 2. To overcome these two challenges and yield a more comprehensive evolutionary history of a B cell lineage, we propose to infer lineage trees that jointly models both sequence evolution (SHM) and isotype evolution (CSR).

Similarly to SHM, one could model the evolution of CSR using unweighted parsimony. That is, one would prefer lineage trees $T$ with isotypes $\beta : V(T) \to [r]$ that minimize the number of isotype changes, i.e., $\sum_{(u,v) \in E(T)} D(\beta(u), \beta(v))$. However, there are two issues with this approach. First, it does not appropriately penalize lineage trees that violate the irreversible property of isotype evolution.[24] Second, it does not account for the fact that given an isotype starting state the probability of transitioning to each of the possible isotype states is not necessarily equal. In fact, knowing these probability distributions is useful for researchers looking to gain basic insight into the patterns and casual factors of class switch recombination.[27] Therefore, we seek to develop an appropriate evolutionary model for CSR that captures the irreversible property of class switching and models preferential isotype class transitions.

We propose a state or tree dependence model[52,53] evolutionary model for CSR, which models the joint probability distribution of a random variable vector under Markov-like assumptions on a given tree. A dependence tree with, sometimes referred to as a state tree, is a tree that defines the conditional independence structure of the random variables associated with the nodes of the tree.[52,53] Simply put, it is a type of Bayesian network, where the underlying directed acyclic graph is a tree that he conditional independence structure of the random variables associated with the nodes of the tree. For each node $u$ in a dependence tree, we associate a random variable. Here, the random variables of interest in this state tree model are the isotypes $\beta(v)$ of each node $v$ in lineage tree $T$. This model is parameterized by a probability distribution over the isotype of the root and isotype transition probabilities. As the root $v_0$ of a lineage tree $T$ is a naive B cell post V(D)J recombination, the isotype $\beta(v_0)$ is always 1 (IgM) and the probability distribution of $\beta(v_0)$ is defined as $\Pr(\beta(v_0) = 1) = 1$ and 0 otherwise. Intuitively, isotype transition probabilities captures the conditional probability of a descendant isotype given the isotype of its parent subject to irreversible isotype evolution. Next, we give a formal definition of isotype transition probabilities.

**Definition 2.** An $r \times r$ matrix $\mathbf{P} = [p_{s,t}]$ is an *isotype transition probability matrix* provided for all isotypes $s, t \in [r]$ it holds that (i) $p_{s,t} \geq 0$, (ii) $p_{s,t} = 0$ if $s > t$, and (iii) $\sum_{t=1}^{r} p_{s,t} = 1$ for all isotypes $s \in [r]$.

We define the joint likelihood $\mathrm{CSR}(T, \beta, \mathbf{P})$ of the observed isotypes $b$ for isotype transition probabilities $b$ and any lineage tree $T$ whose leaves have isotypes $b$ as,

$$\mathrm{CSR}(T, \beta, \mathbf{P}) \;=\; \Pr(b|T, \alpha, \beta, \mathbf{P}) = \Pr(b|T, \beta, \mathbf{P}) = \prod_{(u,v) \in E(T)} p_{\beta(u), \beta(v)}. \tag{Equation 2}$$

Rather than inferring each lineage tree independently, we seek to infer a lineage tree for each of the $k$ clonotypes with shared isotype transition probabilities $\mathbf{P}^*$, first minimizing $\sum_{j=1}^{k} \mathrm{SHM}(T_j^*, \alpha_j^*)$ and then breaking ties by maximizing $\prod_{j=1}^{k} \mathrm{CSR}(T_j^*, \beta_j^*, \mathbf{P}^*)$.

### The TRIBAL algorithm

The input to TRIBAL is a set of $k$ clonotypes with corresponding maximum parsimony forest $\mathcal{T}_j$ for each clonotype $j$. In addition, we are given isotypes $b_j$ labeling the leaves of trees $\mathcal{T}_j$ for each clonotype $j$ (Figures 1C and 2A). Obtaining this input requires a number of preprocessing steps of a scRNA-seq dataset (Figure 1), including (i) BCR assembly and isotype calling of each sequenced cell, (ii) clonotyping or clustering the cells based on a shared germline alleles for both the heavy and light chains, (iii) obtaining an MSA for sequences within a clonotype and (iv) finding a parsimony forest for each MSA of a clonotype. These preprocessing steps are not part of TRIBAL.

TRIBAL is an algorithm to solve the BLFI problem. It consists of an initialization stage followed by alternately optimizing a B cell lineage tree $T_j$ for each clonotype $j$ and then finding the maximum likelihood estimate for the isotype transition probabilities $\mathbf{P}$ shared across the B cell lineage forest.

### Initialization of isotype transition probabilities

For our coordinate ascent approach, we also require an initialization for the isotype transition probabilities (Figure 2A). We set the initial transition probabilities to reflect the observation that under baseline conditions, the probability of a B cell undergoing class switching is lower than the probability of it maintaining its original antibody class..[17,27] Thus, we initialize $\mathbf{P}^{(1)}$ such that $p_{s,s} > p_{s,t}$ for all isotypes $s$ and $t$. Let $\theta \in [0.5, 1]$ be the probability that a B cell does not class switch, i.e., $p_{s,s} = \theta$ for each isotype $s < r$ and

$p_{s,s} = 1$ if $s = r$. We enforce irreversibility such that $p_{s,t} = 0$, if $s > t$. We then initialize the remaining parameters uniformly, i.e., $p_{s,t} = (1 - p_{s,s})/(r - s)$ where $r$ is the total of number isotypes. We conduct multiple restarts, varying $\theta \in [0.5, 1]$ in each restart.

### Optimizing B cell lineage trees given isotype transition probabilities

We have the following key proposition and corollary.

**Proposition 1.** For any tree $T$ labeled by sequences $\alpha$ and refinement $T'$ of $T$, there exists a sequence labeling $\alpha'$ for $T'$ such that $SHM(T, \alpha) = SHM(T', \alpha')$.

**Corollary 1.** Any lineage tree $T'$ that lexicographically optimizes $SHM(T', \alpha')$ and then $CSR(T', \beta', \mathbf{P})$ must be a refinement of some tree $T$ optimizing only $SHM(T, \alpha)$.

*Proof.* The sequencing labeling $\alpha'$ is found by setting $\alpha'(v) = \alpha(v)$ and $\alpha'(v') = \alpha(v)$ during each EXPAND operation. By construction, the new edge $(v, v')$ has $D(\alpha'(v), \alpha'(v')) = 0$ and every original edge maintains its original Hamming distance in $T'$. Therefore, $SHM(T, \alpha) = SHM(T', \alpha')$.

The inference of optimal lineage trees $T_1^{(\ell)}, \ldots, T_k^{(\ell)}$ is conditionally independent given isotype transition probabilities $\mathbf{P}^{(\ell)}$. We therefore focus our discussion on how TRIBAL infers a B cell lineage tree $T_j^{(\ell)}$ for a single clonotype $j$ during iteration $\ell$ given isotype transition probabilities $\mathbf{P}^{(\ell)}$. By Corollary 1, we solve this problem by finding an optimal refinement $T'$ and corresponding isotype labeling $\beta'$ for each tree $T$ in the input set $\mathcal{T}_j^{(\ell)}$ and select the one that maximizes our CSR objective (Figure 2B). Maximizing the log likelihood of $CSR(T, \beta, \mathbf{P})$ is equivalent to maximizing a weighted parsimony criterion. This leads to the following problem statement.

**Problem 2** (Most Parsimonious Tree Refinement (MPTR)). Given a tree $T$ on $n$ leaves, isotypes $b = [b_0, \ldots, b_n]$ and isotype transition probabilities $\mathbf{P}$, find a tree $T'$ with root $v_0'$ and isotype labels $\beta' : V(T') \to [r]$ such that (i) $T'$ is a refinement of $T$, (ii) $\beta'(v_0') = b_0 = 1$, (iii) $\beta'(v_i') = b_i$ for each leaf $v_i' \in \{v_1', \ldots, v_n'\}$ and (iv) $\log CSR(T', \beta', \mathbf{P})$ is maximum.

We prove below that the MPTR problem is NP-hard, which means that it is very unlikely there exists a fast (polynomial-time) algorithm for solving this problem exactly. As such, we solve an instance $(T, b, \mathbf{P})$ of the MPTR problem (Figure S9) using integer linear programming by reducing it to the following graph problem. Given an instance $(T, b, \mathbf{P})$ of the MPTR, we construct a directed graph $G_{T,\mathbf{b}}$, called the expansion graph, with nodes $V(G_{T,\mathbf{b}}) \subseteq V(T) \times [r]$ and edges $E(G_{T,\mathbf{b}})$. At a high level, nodes of $V(G_{T,\mathbf{b}})$ are of the form $(u, s)$ where $u \in V(T)$ is a node of the input tree $T$ and $s \in [r]$ is an isotype state. Formally, we have the following definition.

**Definition 3.** A directed graph $G_{T,\mathbf{b}}$ is an *expansion graph* of a rooted tree $T$ whose leaves are labeled by isotypes $b$ provided $V(G_{T,\mathbf{b}}) = \bigcup_{u \in V(T)} X(u)$ where,

$$X(u) = \begin{cases} \{(u, b_u)\}, & \text{if } u \in L(T), \\ \{(u, s) | s \in \{1, \ldots, \max\{b_v | v \in L(T_u)\}\}\}, & \text{if } u \in V(T) \backslash L(T), \end{cases} \quad \text{(Equation 3)}$$

and $E(G_{T,\mathbf{b}}) = \{((u,s),(v,t)) | (u,v) \in E(T), s \leq t\} \cup \{((u,s),(u,t)) | u \in V(T), s < t\}$.

In the above definition $X(u)$ is the set of nodes of $G_{T,\mathbf{b}}$ corresponding to node $u$ of $T$, accounting for the fact that leaves $u$ of $T$ retain their isotype state in any refinement $T'$ of $T$. On the other hand, internal nodes $u$ of $T$ may be subject to EXPAND operations such that the corresponding nodes of $T'$ are assigned isotypes $s$ ranging from state 1 to the maximum isotype state among all descendant leaves of $u$ in. The edges of $G_{T,\mathbf{b}}$ respect the irreversibility property of isotypes as well as the parental relationships of nodes of $T$. See Figure S9 for an example expansion graph $G_{T,\mathbf{b}}$.

We now define constrained subtrees, termed valid, of the expansion graph $G_{T,\mathbf{b}}$.

**Definition 4.** A subtree $T'$ of $G_{T,\mathbf{b}}$ is *valid* provided (i) $T'$ is rooted at $(v_0, 1)$ where $v_0$ is the root of $T$ and (ii) there is a unique edge $((u, s), (v, t))$ in $E(T')$ for each edge $(u, v)$ of $T$.

We now show that the set of valid subtrees of $G_{T,\mathbf{b}}$ corresponding to trees $T'$ with isotype labelings $\beta'$ is equivalent to the set composed of pairs $(T', \beta')$ where $T'$ is a refinement of $T$ and $\beta'$ is a transitory isotype labeling of $T'$.

**Lemma 1.** Let $T'$ be a refinement of $T$ whose leaves are labeled by isotypes $b$ and let $\beta'$ be an isotype labeling of $T'$. Then, $\beta'$ is transitory if and only if $(T', \beta')$ induces a valid subtree of $G_{T,\mathbf{b}}$.

**Proof.** ($\Rightarrow$) Let $\beta'$ be a transitory isotype labeling of $T'$. We start by showing that $(T', \beta')$ induce a connected subtree of $G_{T,\mathbf{b}}$. First, let $u'$ be a node of $T'$ labeled by isotype $\beta(u')$. We claim that $(u', \beta(u')) \in X(u)$. We distinguish the two cases. First, $u' \in L(T')$. Let $u = \sigma(u')$ be the original leaf node $u$ of $T$. Since $\beta'$ is transitory, we have $\beta(u') = b_{\sigma(u')} = b_u$. Hence, $(u', \beta(u')) \in X(u)$ for each leaf node $u' \in L(T')$. Second, $u' \in V(T') \backslash L(T')$. Let $u = \sigma(u')$ be the original internal node $u$ of $T$. Suppose for a contradiction $(u', \beta'(u')) \in X(u))$. This means that $\beta'(u') > \max\{b_v | v \in L(T_u)\}$. As such, there would be an edge $(u'', v'')$ such that $\beta'(u'') > \beta'(v'')$ where $u''$ is a node in the subtree $T_{u'}'$ rooted at node $u'$. However, this would mean that $\beta'$ would violate condition (iii) of Definition 5, a contradiction. Thus, $(u', \beta(u')) \in X(u)$ for each internal node $u' \in V(T' \backslash L(T'))$. Hence, $(u', \beta(u')) \in V(G_{T,\mathbf{b}})$.

We now prove that each edge $(u', v')$ of $T'$ whose incident nodes are labeled by $(\beta'(u'), \beta'(v'))$ corresponds to an edge $((u', \beta'(u')), (v', \beta'(v')))$ of $G_{T,\mathbf{b}}$. This follows directly from conditions (iii) and (iv) of Definition 5 and the definition of $E(G_{T,\mathbf{b}})$ in Definition 3. This implies that the subgraph of $G_{T,\mathbf{b}}$ induced by $(T', \beta')$ is a (connected) subtree of $G_{T,\mathbf{b}}$.

We now must show that this induced subtree of $G_{T,\mathbf{b}}$ is valid. By condition (i) of Definition 5, we have that $\beta'(v_0') = 1$ for the root $v_0'$ of $T'$. As such, the induced subtree of $G_{T,\mathbf{b}}$ is rooted at $(v_0', 1)$. Finally, we must show there is a unique edge $((u, s), (v, t))$ in the induced subtree of $G_{T,\mathbf{b}}$ for each original edge $(u, v)$ of $T$. This follows from the fact that $T'$ is a refinement of $T'$. Thus the subgraph of $G_{T,\mathbf{b}}$ induced by $(T', \beta')$ is a valid subtree of $G_{T,\mathbf{b}}$.

($\Leftarrow$) Consider a valid subtree of $G_{T,\mathbf{b}}$, resulting in a tree $T'$ and isotype labeling $T'$. To see why $T'$ is a refinement of $T$, observe that edges $((u,s),(u,t))$ correspond to an EXPAND operation on node $u$ of $T$. It remains to show that $\beta'$ is transitory. By condition (i) of Definition 4, we have that the root of $T'$ is labeled by state 1, satisfying condition (i) of Definition 5. Conditions (ii) and (iii) of Definition 5 are met by construction of $G_{T,\mathbf{b}}$. Finally, condition (iv) of Definition 5 follows from condition (ii) of Definition 4. Hence, the isotype labeling $\beta'$ of $T'$ is transitory.

The following key proposition follows from the previous two lemmas.

**Proposition 2.** Let $G_{T,\mathbf{b}}$ be an expansion graph of a rooted tree $T$ whose leaves are labeled by isotypes $b$. Then, given isotype transition probabilities $\mathbf{P}$, a valid subtree $(T',\beta')$ of $G_{T,\mathbf{b}}$ maximizing $\sum_{(u',v')\in E(T')} \log p_{\beta'(u'),\beta'(v')}$ is an optimal solution to MPTR instance $(T, b, \mathbf{P})$.

To find such a valid subtree with maximum log likelihood, we formulate a mixed integer linear program based on a multi-commodity flow formulation for modeling connectivity. We make use of two sets of decision variables. The first is $f^t_{(u,s),(v,t)} \in \mathbb{R}_{\geq 0}$, which represents the amount of flow on edge $(u,v)$ designated for sink $q \in L(T)$. The second is $x_{(u,s),(v,t)} \in \{0,1\}$, which indicates if edge $(u,v)$ has non-zero flow.

$$\min \sum_{((u,s),(v,t))\in E(G_{T,\mathbf{b}})} x_{(u,s),(v,t)} \log p_{s,t} \qquad \text{(Equation 4)}$$

s.t.

$$\sum_{(v,t)\in \eta^+((u,s))} f^q_{(u,s),(v,t)} = \sum_{(v,t)\in \eta^-((u,s))} f^q_{(v,t),(u,s)}, \quad \forall v\in V(T)\backslash L(T)(v,s)\in V(G_{T,\mathbf{b}})\backslash\{(v_0,1)\}, q\in L(T), \qquad \text{(Equation 5)}$$

$$\sum_{(u,s)\in \eta^-((q,b_q))} f^q_{(u,s),(q,b_q)} = 1, \forall q\in L(T), \qquad \text{(Equation 6)}$$

$$\sum_{(v,t)\in \eta^+((v_0,1))} f^q_{(v_0,1),(v,t)} = 1, \forall q\in L(T), \qquad \text{(Equation 7)}$$

$$f^q_{(u,s),(v,t)} \leq x_{(u,s),(v,t)}, \forall q\in L(T), ((u,s),(v,t))\in E(G_{T,\mathbf{b}}), \qquad \text{(Equation 8)}$$

$$\sum_{(u,s)\in X(u)}\sum_{(v,t)\in X(v)} x_{(u,s),(v,t)} = 1, \forall (u,v)\in E(T), \qquad \text{(Equation 9)}$$

$$0 \leq f^q_{(u,s),(v,t)} \leq 1, \forall q\in L(T), ((u,s),(v,t))\in E(G_{T,\mathbf{b}}), \qquad \text{(Equation 10)}$$

$$x_{(u,s),(v,t)} \in \{0,1\}, \forall ((u,s),(v,t))\in E(G_{T,\mathbf{b}}), \qquad \text{(Equation 11)}$$

where $\eta^+((u,s))$ is the set of direct successors of node $(u,s)$ in graph $E(G_{T,\mathbf{b}})$ and $\eta^-((u,s))$ is the set of direct predecessors of node $(u,s)$.

Constraints (5), (6), (7) enforce flow conversation and ensure that each terminal receives one unit of flow. Below is a description of each of the above constraints. Constraint (8) links the flow variables to the choice of edges in the resulting refinement. Finally, constraint (9) ensures that refined tree $T'$ can be obtained from tree $T$ via a series of EXPAND operations.

## Optimizing isotype transition probabilities given B cell lineage trees

Under our state tree model for class switch recombination, we compute the likelihood $\text{CSR}(T,\beta,\mathbf{P})$ for observed isotypes $b$ given a lineage tree $T$ with isotypes $\beta$ and isotype transition probabilities as follows.

$$\begin{aligned}\text{CSR}(T,\beta,\mathbf{P}) &= \Pr(b|T,\beta,\mathbf{P}) \\ &= \prod_{(u,v)\in E(T)} p_{\beta(u),\beta(v)} \\ &= \prod_{v\in V(T)\backslash\{v_0\}}\prod_{(s,t)\in[r]\times[r]} p_{s,t}^{\mathbf{1}(\beta(v)=t,\beta(\phi(v))=s)} \\ &= \prod_{(s,t)\in[r]\times[r]} p_{s,t}^{N_{s,t}}\end{aligned} \qquad \text{(Equation 12)}$$

where $N_{s,t}$ is the count of occurrences in lineage tree $T$ such that $\beta(v) = t$ and $\beta(\phi(v)) = s$. This is easily extended for a set of $k$ lineage trees $T_1,\ldots,T_k$ with corresponding isotypes $\beta_1,\ldots,\beta_k$. Given isotype transition probabilities $P$, the computation of each $\text{CSR}(T_j,\beta_j,\mathbf{P})$

for each clonotype $j$ is conditionally independent, resulting in the joint likelihood,

$$
\begin{aligned}
\prod_{j=1}^{k} \mathrm{CSR}(T_j, \beta_j, \mathbf{P}) \quad &= \prod_{j=1}^{k} \Pr(b_j | T_j, \beta_j, \mathbf{P}) \\
&= \prod_{j=1}^{k} \prod_{(u,v) \in E(T_j)} p_{\beta_j(u), \beta_j(v)} \\
&= \prod_{j=1}^{k} \prod_{v \in V(T_j) \setminus \{v_0\}} \prod_{(s,t) \in [r] \times [r]} p_{s,t}^{\mathbf{1}\left(\beta_j(v) = t, \beta_j(\phi(v)) = s\right)} \\
&= \prod_{(s,t) \in [r] \times [r]} p_{s,t}^{\sum_{j=1}^{k} N_{j,s,t}}
\end{aligned}
$$

(Equation 13)

where $N_{j,s,t}$ is the count of occurrences in lineage tree $T_j$ such that $\beta(v) = t$ and $\beta(\phi(v)) = s$.

To update the isotype transition probabilities $P$ for a given set $T_1, \ldots, T_k$ of lineage trees correspondingly labeled by isotypes $\beta_1, \ldots, \beta_k$, we seek the maximum likelihood estimate,

$$
\mathbf{P}^* = \arg \max_{\mathbf{P}} \prod_{(s,t) \in [r] \times [r]} p_{s,t}^{\sum_{j=1}^{k} N_{j,s,t}}
$$

(Equation 14)

subject to,

$$
\sum_{t \in [r]} p_{s,t} = 1, \forall s \in [r].
$$

(Equation 15)

We solve this constrained optimization problem using Lagrange multipliers $\lambda_s$ for each state $s$. We first take the log of likelihood $\prod_{j=1}^{k} \mathrm{CSR}(T_j, \beta_j, \mathbf{P})$ with respect to isotype transition probabilities $\mathbf{P}$.

$$
\begin{aligned}
\log \prod_{j=1}^{k} \mathrm{CSR}(T_j, \beta_j, \mathbf{P}) &= \log \prod_{(s,t) \in [r] \times [r]} p_{s,t}^{\sum_{j=1}^{k} N_{j,s,t}} \\
&= \sum_{(s,t) \in [r] \times [r]} \left( \sum_{j=1}^{k} N_{j,s,t} \right) \log p_{s,t}.
\end{aligned}
$$

(Equation 16)

To our log likelihood, we add the term $\lambda_s \left( \sum_{s \in [r]} p_{s,t} - 1 \right)$ for each isotype $s$, resulting in new objective,

$$
\mathcal{L}(\mathbf{P}, \lambda_1, \ldots, \lambda_r) = \left[ \sum_{(s,t) \in [r] \times [r]} \left( \sum_{j=1}^{k} N_{j,s,t} \right) \log p_{s,t} + \sum_{s \in [r]} \lambda_s \left( \sum_{t \in [r]} p_{s,t} - 1 \right) \right]
$$

(Equation 17)

Then, we set the partial derivative of $\mathcal{L}(\mathbf{P}, \lambda_1, \ldots, \lambda_r)$ with respect to each parameter $p_{s,t}$ and $\lambda_s$ and solve the resulting system of equations. For each $\lambda_s$, we obtain our constraint,

$$
\begin{aligned}
\frac{\partial \mathcal{L}}{\partial \lambda_s} = 0 &= \left( \sum_{t \in [r]} p_{s,t} - 1 \right) \\
\sum_{t \in [r]} p_{s,t} &= 1
\end{aligned}
$$

(Equation 18)

For each parameter $p_{s,t}$, we set the partial derivative to 0 and solve for $p_{s,t}$ as a function of $\lambda_s$.

$$
\begin{aligned}
\frac{\partial \mathcal{L}}{p_{s,t}} = 0 &= \frac{\sum_{j=1}^{k} N_{j,s,t}}{p_{s,t}} - \lambda_s \\
\lambda_s &= \frac{\sum_{j=1}^{k} N_{j,s,t}}{p_{s,t}} \\
p_{s,t} &= \frac{\sum_{j=1}^{k} N_{j,s,t}}{\lambda_s}
\end{aligned}
$$

Given the constraint (18), we have that,

$$\sum_{t \in [r]} p_{s,t} = \frac{\sum_{t \in [r]} \sum_{j=1}^{k} N_{j,s,t}}{\lambda_s} = 1, \qquad \text{(Equation 19)}$$

and

$$\lambda_s = \sum_{t \in [r]} \sum_{j=1}^{k} N_{j,s,t}.$$

This yields the following maximum likelihood estimate $p_{s,t}^*$,

$$p_{s,t}^* = \frac{\sum_{j=1}^{k} N_{j,s,t}}{\sum_{t \in [r]} \sum_{j=1}^{k} N_{j,s,t}}$$

Lastly, we apply a pseudocount of 1 to all isotype transition probabilities $p_{s,t}$, where $s \leq t$, in order to account for the potential of any unobserved transitions.

$$p_{s,t}^* = \frac{\sum_{j=1}^{k} N_{j,s,t} + 1}{\sum_{t \in [r]} \left( \sum_{j=1}^{k} N_{j,s,t} + 1 \right)}. \qquad \text{(Equation 20)}$$

## B cell lineage forest inference

Recall the B CELL LINEAGE FOREST INFERENCE PROBLEM (BLFI) from the main text, restated below for convenience.

**(Main Text) Problem 1** (B cell Lineage Forest Inference (BLFI)). Given MSAs $\mathbf{A}_1, \ldots, \mathbf{A}_k$ and isotypes $\mathbf{b}_1, \ldots, \mathbf{b}_k$ for $k$ clonotypes, find isotype transition probabilities $\mathbf{P}^*$ for $r$ isotypes and lineage trees $T_1^*, \ldots, T_k^*$ for $(\mathbf{A}_1, \mathbf{b}_1), \ldots, (\mathbf{A}_k, \mathbf{b}_k)$ whose nodes are labeled by sequences $\alpha_1^*, \ldots, \alpha_k^*$ and isotypes $\beta_1^*, \ldots, \beta_k^*$, respectively, such that $\beta_1^*, \ldots, \beta_k^*$ is minimum and then $\prod_{j=1}^{k} \mathrm{CSR}(T_j^*, \beta_j^*, \mathbf{P}^*)$ is maximum.

**Theorem 1.** The BLFI problem is NP-hard even if $k = 1$ and $r = 1$.

We prove that the BLFI problem is NP-hard via a simple reduction from the LARGE PARSIMONY problem[54](Figure S10). Although this problem is well known, we restate it here for completeness.

**Problem 3** (Large Parsimony (LP)). Given a matrix $\mathbf{A} \in \{0,1\}^{n \times m}$, find a rooted tree $T$ whose nodes are labeled by sequences $\alpha : V(T) \to \{0,1\}^m$ such that the $n$ leaves are labeled by the rows of $\mathbf{A}$ and $\sum_{(u,v) \in E(T)} D(\alpha(u), \alpha(v))$ is minimum.

The reduction to BLFI proceeds by using the same MSA $\mathbf{A}$ directly for a single clonotype, i.e., $k = 1$. Additionally, we restrict the number $r$ of isotypes to 1, and set isotypes $b = [1]^n$.

**Lemma 2.** Tree $T$ and node labeling $\alpha$ form an optimal solution to LP instance $\mathbf{A}$ if and only if tree $T$, sequences $\alpha$ and isotypes $\beta$, the isotype transition probabilities $\mathbf{P}$ form an optimal solution to BLFI instance $(\mathbf{A}, \mathbf{b})$.

**Proof.** ($\Rightarrow$) Let tree $T$ and sequence labeling $\alpha$ be an optimal solution to the LP problem. We will show that $T$ and $\alpha$ can be augmented to form an optimal solution to the corresponding BLFI problem. We set $\mathbf{P} = [1]$. We also set $\beta(v) = 1$ for all nodes $v \in T$. We claim that $(T, \alpha, \beta, \mathbf{P})$ form an optimal solution to BLFI. Assume for a contradiction there exists a solution $(T', \alpha', \beta', \mathbf{P}')$ such that $\mathrm{SHM}(T', \alpha') < \mathrm{SHM}(T, \alpha)$, or $\mathrm{SHM}(T', \alpha') = \mathrm{SHM}(T, \alpha)$ and $\mathrm{CSR}(T', \beta', \mathbf{P}') > \mathrm{CSR}(T, \beta, \mathbf{P})$. Clearly, any feasible solution to BLFI must use $\beta(v) = 1$ for all nodes $v$ and $\mathbf{P} = [1]$ as $r = 1$. This means that any feasible solution to BLFI will have a CSR objective value of 1. Therefore, $\mathrm{CSR}(T', \beta', \mathbf{P}') = \mathrm{CSR}(T, \beta, \mathbf{P}) = 1$. Hence, $\mathrm{SHM}(T', \alpha') < \mathrm{SHM}(T, \alpha)$. As can be seen in 1, the SHM objective equals the objective of the LP problem. Therefore, $T'$ and $\alpha'$ have a lower parsimony score than $T$ and $\alpha$, a contradiction.

($\Leftarrow$) Let $(T, \alpha, \beta, \mathbf{P})$ be an optimal solution to BLFI. Again, as the SHM objective equals the objective of the LP problem, it directly follows that $(T, \alpha)$ form an optimal solution to the LP problem instance.

## Combinatorial characterization of the most parsimonious tree refinement problem

Recall the definition of isotype transition probabilities $\mathbf{P}$, the CSR log likelihood for isotypes $b$ of a tree $T$ with nodes labeled by isotypes $\beta$, and the MOST PARSIMONIOUS TREE REFINEMENT problem, provided below for convenience.

Definition 2. An $r \times r$ matrix $\mathbf{P} = [p_{s,t}]$ is an *isotype transition probability matrix* provided for all isotypes $s, t \in [r]$ it holds that (i) $p_{s,t} \geq 0$, (ii) $p_{s,t} = 0$ if $s > t$, and (iii) $\sum_{t=1}^{r} p_{s,t} = 1$ for all isotypes $s \in [r]$.

$$\log \mathrm{CSR}(T, \beta, \mathbf{P}) = \log \prod_{(u,v) \in E(T)} p_{\beta(u), \beta(v)} = \sum_{(u,v) \in E(T)} \log p_{\beta(u), \beta(v)}.$$

**Problem 2** (Most Parsimonious Tree Refinement (MPTR)). Given a tree $T$ on $n$ leaves, isotypes $b = [b_0, \ldots, b_n]$ and isotype transition probabilities $\mathbf{P}$, find a tree $T'$ with root $v_0'$ and isotype labels $\beta' : V(T') \to [r]$ such that (i) $T'$ is a refinement of $T$, (ii) $\beta'(v_0') = b_0 = 1$, (iii) $\beta'(v_i') = b_i$ for each leaf $v_i' \in \{v_1', \ldots, v_n'\}$ and (iv) $\log \mathrm{CSR}(T', \beta', \mathbf{P})$ is maximum.

Let $\sigma$ be a mapping from $V(T')$ to $V(T)$ that reverses all EXPAND operations of each node $u'$ in refinement $T'$ in order to obtain back the node $\sigma(u') = u$ from which it was derived in the original tree $T$. We say that an isotype labeling $\beta' : V(T') \to [r]$ of $T'$ is transitory if along each directed edge $(u', v')$ of $T'$ either the isotype changes or $u'$ and $v'$ correspond to two distinct nodes of $T$. More formally, we have the following definition.

**Definition 5.** Let $T'$ be a refinement of a tree $T$ whose leaves are labeled by isotypes $b$. Then, an isotype labeling $\beta'$ of $T'$ is *transitory* provided (i) $\beta'(v'_0) = 1$ where $v'_0$ is the root of $T'$, (ii) $\beta'(v') = b_{\sigma(v')}$ for each leaf $v' \in L(T')$, (iii) $\beta'(u') \leq \beta(v')$ for each edge $(u', v')$ of $T'$, and (iv) $\beta'(u') = \beta'(v')$ only if $\sigma(u') \neq \sigma(v')$ for each edge $(u', v')$ of $T'$.

Importantly, among the set of optimal solutions $(T', \beta')$ to each MPTR problem instance $(T, b, \mathbf{P})$ there exist solutions where $\beta'$ is transitory.

**Lemma 3.** Let $(T, b, \mathbf{P})$ be an MPTR problem instance. There exist an optimal solution $(T', \beta')$ where $\beta'$ is transitory.

*Proof.* We prove this by contradiction. Let $(T', \beta')$ be an optimal solution where $\beta'$ is not transitory. First, observe that it holds that $\beta'(u') \leq \beta(v')$ for each edge $(u', v')$ of $T'$. To see why, if there were an edge $(u', v')$ such that $\beta'(u') > \beta(v')$ then $\text{CSR}(T', \beta', \mathbf{P}) = -\infty$ as $\log p_{s,t} = -\infty$ if $s > t$. However, setting $\beta'(u') = 1$ for nodes $\text{CSR}(T', \beta', \mathbf{P}) = -\infty$ not in $L(T')$ would result in log likelihood greater than $-\infty$. Since $(T', \beta')$ is a feasible solution to MPTR respecting irreversibility of isotype transitions, it means that condition (iv) of Definition 5 is violated. Let $(u', v')$ be an edge such that $\beta'(u') = \beta'(v')$ and $\sigma(u') = \sigma(v')$. We can contract this edge, retaining the isotype labeling $\beta'$ for the remaining nodes, such that the resulting tree remains a refinement of $T$ and the objective value remains unchanged as $\log p_{s,s} = 0$. Repeating this procedure for all edges $(u', v')$ such that $\beta'(u') = \beta'(v')$ and $\sigma(u') = \sigma(v')$ results in $(T'', \beta'')$, where $T''$ is a refinement of $T$ labeled by $\beta''$, with the same optimal score as $(T', \beta')$. Clearly, $(T'', \beta'')$ is transitory, proving the lemma.

Complexity of the most parsimonious tree refinement problem.

Note that maximizing the CSR log likelihood is equivalent to maximizing the CSR likelihood, which is the objective function we will use in this subjection. That is,

$$\text{CSR}(T, \beta, \mathbf{P}) = \prod_{(u,v) \in E(T)} p_{\beta(u), \beta(v)}.$$

We now prove the following theorem.

**Theorem 2.** The MPTR problem is NP-hard.

We show that MPTR is NP-hard by reduction from SET COVER.

**Problem 4** (Set Cover). Given a universe $\mathcal{U}$ of elements $\{u_1, \ldots u_{|\mathcal{U}|}\}$ and a collection $S$ of subsets $\{S_1, \ldots, S_{|S|}\}$ such that $\bigcup_{i=1}^{|S|} S_i = \mathcal{U}$, find a cover $C \subseteq S$ such that $\bigcup_{S \in C} S = \mathcal{U}$ and the size $|C|$ of the cover is minimum.

Note that while the order of the subsets in collection $S$ does not matter for SET COVER, our reduction will assume the subsets to be in an arbitrary but fixed order. Similarly, we will assume $\mathcal{U}$ to be ordered arbitrarily. SET COVER has been proven to be NP-hard in Karp's 21 NP-complete problems.[55] We describe a polynomial time reduction from SET COVER to MPTR. To that end, given the set $\mathcal{U}$ of elements and the collection $S$ of subsets, we construct a tree $T$ with $|\mathcal{U}| + 1$ leaves, $r = |\mathcal{U}| + |S| + 2$ isotypes, observed isotypes $b \in [r]^{|\mathcal{U}|+1}$, and $r \times r$ transition probabilities $\mathbf{P}$. The steps are as follows.

(1) To construct tree $T$, we begin by adding the root node $v_0$. Following that, we attach two children, denoted as $\overline{v}_0$ and $v_{|\mathcal{U}|+1}$, to the root node $v_0$. Finally, for each element $u_q \in \mathcal{U}$, we add an edge $(\overline{v}_0, v_q)$ in tree $(\overline{v}_0, v_q)$. The constructed tree $T$ has $|\mathcal{U}| + 3$ nodes and $|\mathcal{U}| + 2$ edges.

(2) We consider a total of $r = |S| + |\mathcal{U}| + 2$ isotypes, each corresponding to either a subset $S_i \in S$, an element $u_q \in \mathcal{U}$, or one of the special symbols $\top$ or $\bot$. Specifically, the first isotype stands for the special symbol $\top$, followed by $|S|$ isotypes representing each subset $S_i \in S$, succeeded by $|\mathcal{U}|$ isotypes representing each element $u_q \in \mathcal{U}$, and concluding with the last isotype signifying the special symbol $\bot$. For convenience, we define a function $R : S \cup \mathcal{U} \cup \{\top, \bot\} \to [r]$ to map the subsets $S_i \in S$, the elements $u_q \in \mathcal{U}$, and the special symbols $\top$ and $\bot$ to their representative isotype indices as follows.

$$\bot.R(X) = \begin{cases} 1, & \text{if } X = \top, \\ i + 1, & \text{if } X = S_i, \\ |S| + q + 1, & \text{if } X = u_q, \\ |S| + |\mathcal{U}| + 2, & \text{if } X = \end{cases}$$

(3) For the observed isotypes, we set $b_0 = b_{|\mathcal{U}|+1} = R(\top) = 1$, and $b_0 = b_{|\mathcal{U}|+1} = R(\top) = 1$ for U1 $\leq q \leq ||$.

(4) We define $\epsilon$ to be a constant such that $0 < \epsilon \leq 1/(|S| + |\mathcal{U}| + 1)$. Next, we construct the isotype transition probabilities $\mathbf{P}$ parameterized by $\epsilon$ as follows.

   (a) We set the transition probability from $R(\top)$ to $R(\top)$ or $R(S_i)$ for any set $S_i \in S$ to be $\epsilon$ and to $R(u_q)$ for any $u_q \in \mathcal{U}$ to be 0.

$$\begin{aligned} p_{R(\top), R(\top)} &= \epsilon, \\ p_{R(\top), R(S_i)} &= \epsilon \quad \forall 1 \leq i \leq |S|, \\ p_{R(\top), R(u_q)} &= 0 \quad \forall 1 \leq q \leq |u| \end{aligned}$$

(b) We set the transition probability from $R(\top)$ to $R(\bot)$ to be $1 - (1 + |S|)\epsilon$.

$$p_{R(\top),R(\bot)} = 1 - (1 + |S|)\epsilon.$$

(1) We set the transition probability $p_{R(S_i),R(S_j)}$ for any $S_i, S_j \in \mathcal{S}$ to be $\epsilon$ if $i < j$, and 0 otherwise.

$$p_{R(S_i),R(S_j)} = \begin{cases} \epsilon, & \text{if } i < j, \\ 0, & \text{if } i \geq j, \end{cases} \forall 1 \leq i, j \leq |s|.$$

(d) We set the transition probability from $R(S_i)$ to $R(u_q)$ for any set $S_i \in \mathcal{S}$ and any element $u_q \in \mathcal{U}$ to be $\epsilon$ if $u_q \in S_i$, and 0 otherwise.

$$p_{R(S_i),R(u_q)} = \begin{cases} \epsilon, & \text{if } u_q \in S_i, \\ 0, & \text{if } u_q \notin S_i, \end{cases} \forall 1 \leq i \leq |S|, 1 \leq q \leq |u|.$$

(e) For each $S_i \in \mathcal{S}$, we set the transition probability from $R(S_i)$ to $R(\top)$ to be 0 and to $R(\bot)$ to be $1 - (|S| - i + |S_i|)\epsilon$.

$$p_{R(S_i),R(\top)} = 0 \qquad 1 \leq i \leq |S|,$$
$$p_{R(S_i),R(\bot)} = 1 - (|S| - i + |S_i|)\epsilon \quad 1 \leq i \leq |s|.$$

(f) For any $u_q \in \mathcal{U}$, we set the transition probability from $R(u_q)$ to any other isotype except $\bot$ to be 0. We set $p_{R(u_q),R(\bot)}$ for any $u_q \in \mathcal{U}$ to be 1.

$$p_{R(u_q),R(X)} = 0 \quad \forall 1 \leq q \leq |\mathcal{U}|, X \in \mathcal{S} \cup \mathcal{U} \cup \{\top\},$$
$$p_{R(u_q),R(\bot)} = 1 \qquad \forall 1 \leq q \leq |u|.$$

(g) Last, we set the transition probability $p_{R(\bot),R(\bot)}$ to be 1.

$$p_{R(\bot),R(\bot)} = 1$$

Clearly, by construction matrix $\mathbf{P}$ obtained from a Set Cover instance $(\mathcal{U}, \mathcal{S})$ is an isotype transition probability matrix as $\mathbf{P}$ is upper triangular, each entry is non-negative and each row sums to 1. In addition, this reduction takes polynomial time.

To prove hardness, let $(T', \beta')$ be an optimal solution to the MPTR instance composed of the input tree $T$, observed isotypes $b$, and isotype transition probabilities $\mathbf{P}$ corresponding to Set Cover instance $(\mathcal{U}, \mathcal{S})$.

**Lemma 4.** $\mathrm{CSR}(T', \beta', \mathbf{P}) > 0$ for the refined tree $T'$ and the isotype labeling $\beta'$ inferred by MPTR.

*Proof.* We prove this by showing that for any constructed input tree $T$, observed isotypes $b$ and isotype transition probabilities $\mathbf{P}$, there exists a refined tree $T'$ and isotype labeling $\beta'$ such that $\mathrm{CSR}(T', \beta', \mathbf{P}) > 0$. We provide a proof by constructing a refined tree $T'$ with isotype labeling $\beta'$. The tree $T'$ will expand the unique polytomous node $\bar{v}_0$ into a chain $\bar{v}_1 \to \ldots \to \bar{v}_{|S|}$. We leave the remaining nodes $v_0, v_1, \ldots, v_{|\mathcal{U}|+1}$ of $T$ unaltered, letting $v'_0, v'_1, \ldots, v'_{|\mathcal{U}|+1}$ denote their corresponding nodes in $T'$. Next, for each $1 \leq q \leq |\mathcal{U}|$, we pick a subset $S_i$ such that $u_q \in S_i$, and add edge $(\bar{v}'_i, v'_q)$ in $T'$ and set $\beta'(v'_q) = R(u_q)$. We add the edges $(v'_0, v'_{|\mathcal{U}|+1})$ and $(v'_0, \bar{v}'_1)$. Finally, we set $\beta'(v'_0) = \beta'(v'_{|\mathcal{U}|+1}) = R(\top)$. Clearly all the edges in $T'$ have nonzero isotype transition probabilities, so $\mathrm{CSR}(T', \beta', \mathbf{P}) > 0$.

**Corollary 2.** The root $v'_0$ of $T'$ is labeled by isotype $\top$.

*Proof.* Due to the presence of leaf $v_{|\mathcal{U}|+1}$ with isotype $b_{|\mathcal{U}|+1} = R(\top)$, the root $v'_0$ of $T'$ must be labeled by isotype $\beta'(v'_0) = R(\top)$, otherwise there would be a zero-probability edge. □

**Corollary 3.** No node $v'$ of $T'$ is labeled by isotype $\bot$.

**Corollary 4.** Each edge $(v', v'')$ of $T'$ has an isotype transition probability of $p_{\beta'(v'),\beta'(v'')} = \epsilon$.

Observe that $\bar{v}_0$ is the only polytomous node in $T$. We will now prove that $\bar{v}_0$ is the only node of $T$ that is expanded in the refined tree $T'$.

**Lemma 5.** Node $\bar{v}_0$ is the only node of $T$ that is expanded in $T'$.

*Proof.* By Lemma 3, we may assume that $\beta'$ is transitory. Let $v'_0$ be the root of $T'$. We prove this lemma by contradiction. Let $v \neq \bar{v}_0$ be a distinct node of $T$ that is expanded in $T'$. We distinguish the following three cases.

- $v = v_{|\mathcal{U}|+1}$: In this case, $v$ equals the leaf node $v_{|\mathcal{U}|+1}$ whose parent is the root $v_0$. Consider the corresponding node $v'_{|\mathcal{U}|+1}$ of $T'$ such that $\sigma(v'_{|\mathcal{U}|+1}) = v_{|\mathcal{U}|+1}$ and $v'_{|\mathcal{U}|+1}$ is a leaf of $T'$. Since $\beta'$ is transitory, we have that $\beta'(v_0) = \beta'(v'_{|\mathcal{U}|+1}) = R(\top)$. Since node $v_{|\mathcal{U}|+1}$ was expanded, node $v'_{|\mathcal{U}|+1}$ has a unique parent $v''_{|\mathcal{U}|+1} \neq v'_0$. As $\beta'$ is transitory and $\beta'(v'_{|\mathcal{U}|+1}) = R(\top)$ and $R(\top) \leq s$ for all $s \in [r]$, we must have that $\beta'(v''_{|\mathcal{U}|+1}) = R(\top)$. This, however, implies that $\beta'$ is not transitory as $\sigma(v''_{|\mathcal{U}|+1}) = \sigma(v'_{|\mathcal{U}|+1}) = v_{|\mathcal{U}|+1}$ and $\beta'(v''_{|\mathcal{U}|+1}) = \beta'(v'_{|\mathcal{U}|+1}) = R(\top)$, which yields a contradiction.

- $v \in \{v_1, \ldots, v_{|\mathcal{U}|}\}$: Note that $v$ is a leaf of $T$. Consider the corresponding node $v'$ of $T'$ such that $T'$ and $v'$ is a leaf of $T'$. The parent of $v$ in $T$ is node $\overline{v}_0$. Since node $v$ was expanded, node $v'$ has a unique parent $v''$ such that $\sigma(v'') = v$. Let $v'''$ be the unique parent of $v''$. By Corollary 4, we have that the two edges $(v'', v')$ and $(v''', v'')$ both have probabilities $\epsilon$, contributing a factor of $2\epsilon$ to the overall probability $\text{CSR}(T', \beta', \mathbf{P})$. However, by contracting the edge $(v'', v')$ and removing the node $v''$, we obtain another solution with higher probability, leading to a contradiction.

- $v = v_0$: Consider the corresponding node $v'_0$ such that $\sigma(v'_0) = v_0$ and $v'_0$ is the root of $T'$. There are two cases two consider. Let $v''_0$ be a child of $v'_0$ such that $\sigma(v''_0) = v_0$. We distinguish two cases.

– First, $\beta'(v'_0) = \beta'(v''_0)$. By Corollary 2, we have that $\beta'(v'_0) = \beta'(v''_0) = R(\top)$. By Corollary 4, we have that the edge $(v'_0, v''_0)$ contributes a factor of $\epsilon$ to the overall probability $\text{CSR}(T', \beta', \mathbf{P})$. We can remove this factor by simply contracting the edge $(v'_0, v''_0)$, resulting in a more optimal solution, which is a contradiction.

– Second, $\beta'(v'_0) \neq \beta'(v''_0)$. By Corollary 2, we have that $\beta'(v'_0) = R(\top)$. By Lemma 4, we have $\beta'(v''_0) \in \{R(S_1), \ldots, R(S_{|S|})\}$. Again, by the same lemma, all children of $v''_0$ will be labeled by isotypes different than $v''_0$. In particular, each child of $v''_0$ will either correspond to node $v_0$ or $\overline{v}_0$ of $T$, labeled from the set $\{R(S_1), \ldots, R(S_{|S|})\} \setminus \{\beta'(v''_0)\}$. Thus, we may contract the edge $(v'_0, v''_0)$, with probability $\epsilon$, and remove the node $v''_0$, reassigning all children of $v''_0$ to $v'_0$. The resulting tree and isotype labeling will have a larger probability, a contradiction.

Assume that a series of EXPAND operations on $\overline{v}_0$ in $T$ has generated $k$ nodes in $T'$, where $k$ ranges from 1 (no EXPAND operation) to $|\mathcal{U}|$. We denote $\overline{v}'_1, \ldots, \overline{v}'_k$ to be the new nodes in $T'$ originating from $\overline{v}_0$ in $T$, i.e., $\sigma(\overline{v}'_1) = \ldots = \sigma(\overline{v}'_k) = \overline{v}_0$. Let $\overline{T}'$ be the subtree of $T'$ induced by nodes $\overline{v}'_1, \ldots, \overline{v}'_k$.

**Lemma 6.** The refined tree $T'$ has $|\mathcal{U}| + k + 2$ nodes, $|\mathcal{U}| + k + 1$ edges, and $\text{CSR}(T', \beta', \mathbf{P}) = \epsilon^{|\mathcal{U}|+k+1}$.

*Proof.* Since $T$ has $|\mathcal{U}| + 3$ nodes, and, by Lemma 5, the only node $\overline{v}_0$ of $T$ that is expanded, expands to $k$ nodes $\overline{v}'_1, \ldots, \overline{v}'_k \in V(T')$, the total number of nodes in $T'$ is $|\mathcal{U}| + 2 - 1 + k = |\mathcal{U}| + k + 2$. Similarly, the number of edges in $T$ is $|\mathcal{U}| + 2$, and since $\overline{T}'$ is a tree containing $k$ nodes, it has $k - 1$ edges. So the total number of edges in $T'$ is $|\mathcal{U}| + 2 + k - 1 = |\mathcal{U}| + k + 1$. It follows from Corollary 4 that $\text{CSR}(T', \beta', \mathbf{P}) = \epsilon^{|\mathcal{U}|+k+1}$.

**Lemma 7.** Nodes $\overline{v}'_1, \ldots, \overline{v}'_k$ of $T'$ are labeled by $k$ distinct isotypes from the set $\{R(S_1), \ldots, R(S_{|S|})\}$.

*Proof.* By construction of $\mathbf{P}$, $R(u_q)$ can only be transitioned into from $R(S_i)$ with nonzero probability where $u_q \in S_i$. So if there is an edge $(\overline{v}'_j, v'_q)$ in $T'$ connecting expanded node $\overline{v}'_j$ with leaf $v'_q$ labeled with $R(u'_q)$ then $\beta'(\overline{v}'_j) = S_i$ for some $S_i \in S$. Using the observation, we begin by showing that each expanded node $\overline{v}'_i$ has at least one child $v'_q \in L(T')$. We do so by contradiction. Suppose the refined tree $T'$ has an expanded node $\overline{v}'_i$ that does not have any leaf $v'_q \in L(T')$ as a child. Without loss of generality, assume that $\overline{v}'_i$ has a child $\overline{v}''_i$, which, in turn, is the parent of a leaf $v'_q \in L(T')$. This means that $\overline{v}''_i$ is labeled with $\beta'(\overline{v}''_i) = R(S_i)$ for some $S_i \in S$. Since $R(S_i)$ can only be transitioned into from $R(S_j)$, where $j < i$, or $R(\top)$ with nonzero probability, it holds that $\beta(\overline{v}'_i)$ is either $R(S_j)$ where $j < i$ or $R(\top)$. Similarly, the parent of $\overline{v}'_i$ should also be labeled either with $R(S_{j'})$ where $j' < j$ or $R(\top)$. Now we create a new tree $T''$ by (i) adding the children of $\overline{v}'_i$ as the children of the parent of $\overline{v}'_i$, and (ii) deleting the edge between $\overline{v}'_i$ and its parent. Clearly $T''$ has nonzero transition probabilities on all the edges, but has one fewer edge than $T'$. So $\text{CSR}(T'', \beta', \mathbf{P}) < \text{CSR}(T', \beta', \mathbf{P})$, which contradicts with the premise that $T'$ minimizes $\text{CSR}(T', \beta', \mathbf{P})$. So each expanded node $\overline{v}'_j$ is labeled with $R(S_i)$ for some $S_i \in S$.

It remains to show that the $k$ nodes $\overline{v}'_1, \ldots, \overline{v}'_k$ are labeled by $k$ distinct isotypes from the set $\{R(S_1), \ldots, R(S_{|S|})\}$. To see why, observe that, by construction of $\mathbf{P}$, the incident nodes of each edge among nodes $\overline{v}'_1, \ldots, \overline{v}'_k$ must be labeled by distinct isotypes from the set $\{R(S_1), \ldots, R(S_{|S|})\}$, as $p_{R(S_i), R(S_i)} = 0$ for all $S_i \in S$.

**Lemma 8.** There exists an minimum set cover of size $k$ if and only if there is an optimal solution $(T', \beta')$ such that $\text{CSR}(T', \beta', \mathbf{P}) = \epsilon^{|\mathcal{U}|+k+1}$.

*Proof.* ($\Rightarrow$) Let $C = \{S_1^*, \ldots, S_k^*\}$ be a set cover of minimum size $k$. Without loss of generality, we further assume that $R(S_i^*) < R(S_{i+1}^*)$ for any $1 \leq i \leq k - 1$. Next, we build a refined tree $T'$ with isotype labeling $\beta'$ by expanding the node $\overline{v}_0 \in V(T)$ to $k$ nodes $\overline{v}'_1, \ldots, \overline{v}'_k \in V(T')$. More specifically, we replace $\overline{v}_0$ with $\overline{v}'_1, \ldots, \overline{v}'_k \in V(T')$ such that (i) $v_0$ is connected to $\overline{v}'_1$ by an edge, (ii) there is an edge $(\overline{v}'_i, \overline{v}'_{i+1})$ in $T'$ for each $1 \leq i \leq k - 1$, (iii) $\overline{v}'_i$ is labeled with $R(S_i^*)$, i.e., $\beta'(\overline{v}'_i) = R(S_i^*)$, and (iv) for each child $v_q$ of $\overline{v}_0$ in $T$, there exists exactly one edge $(\overline{v}'_i, v_q)$ in $T'$ where $u_q \in S_i^*$. Clearly $T'$ is a refinement of tree $T$, and all the newly added edges have nonzero transition probabilities $\epsilon$. Hence, $\text{CSR}(T', \beta', \mathbf{P}) = \epsilon^{|\mathcal{U}|+k+1}$.

All that remains to show is that $(T', \beta')$ is optimal. We show this by contradiction. Let $(T'', \beta'')$ be an optimal solution such that $\text{CSR}(T'', \beta'', \mathbf{P}) < \text{CSR}(T', \beta', \mathbf{P}) = \epsilon^{|\mathcal{U}|+k+1}$. By Lemma 5, we have that only the node $\overline{v}_0$ of $T$ is expanded in $T''$ corresponding $\overline{v}''_1, \ldots, \overline{v}''_{k'}$ nodes in $T''$. Since $\text{CSR}(T'', \beta'', \mathbf{P}) < \text{CSR}(T', \beta', \mathbf{P})$, it must hold that $k'' < k$. By Lemma 7 we have that the $k'$ labels of nodes $\overline{v}''_1, \ldots, \overline{v}''_{k'}$ correspond to $k'$ distinct subsets of $S$. By Lemma 4, we have that these $k'$ subsets of $S$ form a cover of the universe $\mathcal{U}$, leading to a contradiction. Hence, $(T', \beta')$ is optimal.

($\Leftarrow$) Now assume that there exists an optimal solution $(T', \beta')$ such that $\text{CSR}(T', \beta', \mathbf{P}) = \epsilon^{|\mathcal{U}|+k+1}$. Note that the restriction that $\text{CSR}(T', \beta', \mathbf{P}) = \epsilon^{|\mathcal{U}|+k+1}$ is without loss of generality due to Lemma 6. Now according to Lemma 7, there are $k$ expanded nodes in

$T'$ labeled with $R(S_1^*),...,R(S_k^*)$. We define $C = \{S_1^*,...,S_k^*\}$. Now each leaf $v_q' \in L(T')$ labeled with $R(u_q)$ is the child of an expanded node $\overline{v}_i' \in V(T')$ labeled with $R(S_i^*)$. Since $CSR(T', \beta', \mathbf{P}) > 0$ by Lemma 4, the transition probability from $CSR(T', \beta', \mathbf{P}) > 0$ to $R(u_q)$ is strictly greater than 0, which means $u_q \in S_i^*$. So every element in $\mathcal{U}$ is covered by one of the subsets from $C$. So $C$ is a set cover of size $k$.

It remains to show that $C$ is a minimum-size set cover. Assume for a contradiction that there exists a cover $C' = \subseteq S$ such that $|C'| = k' < k = |C|$. Let $C' = \{C_1',...,C_{k'}'\}$ where the subsets follow the same order as in the original reduction to MPTR. We construct a refined tree $T''$ with isotype labeling $\beta''$ corresponding to $C'$ by expanding the unique polytomous node $\overline{v}_0$ of $T$ into a chain $\overline{v}_1'' \to ... \to \overline{v}_{k'}''$, with one node $\overline{v}_i''$ for each subset $C_i \in C'$ labeled by $\beta''(\overline{v}_i'') = R(C_i)$, and connecting each leaf $v_q \in \{v_1,...,v_{|\mathcal{U}|}\}$ to a single expanded node $\overline{v}_i''$ such that $u_q \in C_i'$. Since $C'$ is a cover of $\mathcal{U}$, each leaf $v_q \in \{v_1,...,v_{|\mathcal{U}|}\}$ will be connected. Moreover, tree $T''$ with isotype labeling $\beta''$ form a solution to MPTR. Clearly, $T''$ has $|\mathcal{U}| + k' + 2$ nodes and $|\mathcal{U}| + k' + 1$ edges. Moreover, each edge of $T''$ has a nonzero isotype transition probability equal to $\epsilon$, so $CSR(T'', \beta'', \mathbf{P}) = \epsilon^{|\mathcal{U}|+k'+1} < \epsilon^{|\mathcal{U}|+k+1} = CSR(T', \beta', \mathbf{P})$, a contradiction.

## Simulation details

We designed *in silico* experiments to evaluate TRIBAL with known ground-truth isotype transition probabilities $\mathbf{P}$ and lineage trees $T$ labeled by sequences $\alpha$ and isotypes $\beta$. Specifically, we used an existing BCR phylogenetic simulator[24] that models SHM) but not CSR. We generated isotype transition probabilities $\mathbf{P}$ with $r = 7$ isotypes (as in mice) under two different models of CSR. Briefly, both CSR models assume the probability of not transitioning is higher than the probability of transitioning, but in the *sequential model* there is clear preference for transitions to the next contiguous isotype, while in the *direct model* the probabilities of contiguous and non-contiguous class are similar (Figure S11). Given $\mathbf{P}$, we evolved isotype characters down each ground truth lineage tree $T$.

We generated 5 replications of each CSR model for $k = 75$ clonotypes and $n \in \{35, 65\}$ cells per clonotype, resulting in 20 *in silico* experiments, yielding a total of 1500 ground truth lineage trees. We generated our *in silico* experiments to evaluate all aspects of TRIBAL while benchmarking against existing methods including dnapars,[20] dnaml[20] and IgPhyML.[8]

### SHM simulation and benchmarking

The Davidsen and Matsen SHM simulator models the generation of B cell lineage trees via a Poisson branching process with selection toward BCRs with increased affinity.[24] We used the provided Docker Hub image container (krdav/bcr-phylo-benchmark) to generate our ground truth B cell lineage trees $T$ and sequence labels $\alpha$. In addition, we used the provided benchmarking pipeline to run dnapars,[20] dnaml[20] and IgPhyML.[8] Below is the command to generate our *in silico* experiments for $n \in \{35, 65\}$ cells and $k = 75$ clonotypes and run comparison methods.

```
simulate
  --igphyml
  --dnapars
  --dnaml
  --selection
  --target_dist=5
  --target_count=100
  --carry_cap=1000
  --T=35
  --lambda=2.0
  --lambda0=0.365
  --n={n}
  --nsim={k}
  --random_naive=sequence_data/AbPair_naive_seqs.fa
```

### CSR simulation

After generating each ground truth B cell lineage tree $T$ as described above, we then evolved isotype characters down each tree $T$ using two different models for class switch recombination to obtain ground truth isotypes $\beta$. First, we describe the two different CSR models that we used to generate ground truth isotype transition probabilities $\mathbf{P}$. Then, we describe the generation of these isotype transition probability matrices under these two models.

We grouped each isotype transition probability $p_{s,t}$ where $s \leq t$ into one of three categories: (i) *stay*, (ii) *next*, and (iii) *jump* (Figure S11). In *stay*, the B cell does not undergo any class switching and the isotype does not change. In *next*, a B cell class switches to the next contiguous heavy chain locus. In *jump*, the B cell class switches by jumping to an isotype heavy chain constant locus that is not contiguous.

Next, we describe how we generated ground truth isotype transition probabilities **P** under both direct and sequential CSR models. To simulate isotype transition probabilities with direct switching, we randomly sampled a probability of transitioning $1 - \theta \in \{0.1, 0.15, \ldots, 0.35\}$. We then set the initial isotype transition probabilities as,

$$p'_{s,t} = \begin{cases} 0, & \text{if } s > t, \\ \min(\theta + \epsilon, \tau), & \text{if } s = t, \\ \min\left(\dfrac{1 - \theta}{r - s} + \epsilon, \tau\right), & \text{if } s < t, \end{cases}$$

where we add Gaussian noise $\epsilon \sim \mathcal{N}(\mu, \sigma)$ with mean $\mu = 0.05$ and standard deviation $\sigma = 0.025$ to each parameter. To avoid negative transition probabilities we set $\tau = 0.01$. Figure S11 shows an example of a simulated isotype transition probability matrix under the direct CSR model.

$$p'_{s,t} = \begin{cases} 0, & \text{if } s > t, \\ \min(\theta + \epsilon, \tau), & \text{if } s = t, \\ \min(1 - \theta + \epsilon, \tau), & \text{if } t = s + 1, \\ \tau, & \text{otherwise.} \end{cases}$$

We then set parameter $p'_{s,t} := p'_{s,t} / \sum_{s \in [r]} p'_{s,t}$ to ensure each row in the isotype transition probability matrix $P$ sums to 1. Figure S11 shows an example of a simulated isotype transition probability matrix under the direct model. Figure S11 shows an example of a simulated isotype transition probability matrix under the sequential CSR model.

### Inference using TRIBAL

We ran TRIBAL in two ways, referred to as TRIBAL and TRIBAL-No Refinement, in order to assess the importance of the tree refinement stage of our algorithm. As the naming convention implies, the main difference between TRIBAL and TRIBAL-No Refinement, is that in TRIBAL-No Refinement the input trees are not refined and the isotypes $\widehat{\beta}$ are inferred using the Sankoff[56] algorithm with weights $w_{s,t}^{(\ell)} = -\log p_{s,t}^{(\ell)}$. All other steps of TRIBAL algorithm remain the same.

Due to large input sets $\mathcal{T}_j$ for some simulated clonotypes $j$, we sample 50 trees from $\mathcal{T}_j$ for consideration of candidate lineage tree $T_j^{(\ell)}$ within each iteration $\ell$. We additionally include the previous optimal lineage tree $\ell$ of iteration $\ell - 1$ in the sampled trees for each clonotype $j$ to ensure convergence.

We used a convergence threshold of 0.5 and a maximum of 10 iterations per restart. A total of 5 restarts were performed by iterating through $\theta \in \{0.55, 0.65, 0.75, 0.85, 0.95\}$ for each restart.

### *In silico* study performance metrics
### *Kullback-Leibler (KL) divergence*

To evaluate accuracy of isotype transition probability inference, we used *Kullback–Leibler (KL) divergence*[30] to compare the inferred transition probability distribution $\widehat{p}_s$ of each isotype $s$ to the simulated ground truth distribution $p_s$. KL divergence $D_{\text{KL}}$ is defined as,

$$D_{\text{KL}}(\widehat{p}_s \| p_s) = \sum_{q \in [r]} \widehat{p}_{s,t} \log(\widehat{p}_{s,t} / p_{s,t}) \qquad \text{(Equation 21)}$$

The lower the KL divergence, the more similar the two distributions. To assess accuracy of lineage tree inference, we used normalized Robinson-Foulds (RF) distance to assess accuracy of the topology of the inferred tree $\widehat{T}$, most recent common ancestor (MRCA) distance to assess accuracy of the inferred sequences $\widehat{\alpha}$, and Class Switch Recombination (CSR) error to assess accuracy of the inferred isotypes $\widehat{\beta}$.

### *Normalized Robinson-Foulds (RF) distance*

To assess the accuracy of topology of the inferred B cell lineage tree $\widehat{T}$ with respect to simulated ground truth tree $T$, we used *normalized Robinson-Foulds (RF) distance*. For this metric, we treat both trees as unrooted. For an unrooted tree, if you remove an edge (but not its endpoints), it defines a bipartition of the leaf set.[57] Doing this for every edge in tree $T$ yields a set $B(T)$ of bipartitions. RF distance is defined as the size of the symmetric difference between bipartitions $B(T)$ and $B(\widehat{T})$[31]. We then normalize this by the total number of bipartitions in each tree. Thus, normalized RF is computed as follows,

$$\text{normalizedRF}(T, \widehat{T}) = \frac{|B(T) \Delta B(\widehat{T})|}{|B(T)| + |B(\widehat{T})|}. \qquad \text{(Equation 22)}$$

### *Most recent common ancestor (MRCA) distance*

To assess the accuracy of the inferred ancestral sequence reconstruction $\widehat{\alpha}$ with respect to simulated ground truth $\alpha$, we used a metric called *Most Recent Common Ancestor (MRCA) distance* introduced by Davidsen and Matsen.[24] For any two simulated B cells (leaves), the MRCA distance is the Hamming distance between the MRCA sequences of these two B cells in both the ground truth and inferred lineage trees. This distance is then averaged over all pairs of simulated B cells. A graphical depiction of this metric is show in Figure S12A.

More formally,

$$\text{MRCAdistance}(\alpha, \widehat{\alpha}) = \frac{2}{n(n-1)m} \sum_{u,v \in L(T)} D(\widehat{\alpha}(\widehat{T}, u, v), \alpha(T, u, v)), \tag{Equation 23}$$

where in a slight abuse of notation $\alpha(T, u, v)$ is the sequence of the most recent common ancestor (MRCA) of nodes $u$ and $v$ in lineage tree $T$ and $m$ is the length of MSA.

### Class switch recombination (CSR) error

We assessed the accuracy of isotype inference by a new metric called *CSR error*, which is computed for each B cell $i$ and clonotype $j$ and is the absolute difference between the number of ground-truth class switches and inferred number of class switches that occurred along its evolutionary path from the root (Figure S12B). Since dnaml, dnapars and IgPhyML do not infer isotypes for internal nodes, we pair these methods with the Sankoff algorithm[56] using $w_{s,t}$ equals 1 if $s = t$, 0 if $s < t$ and $\infty$ otherwise.

### Average clade entropy for a leaf labeling

We describe a metric used to assess the average entropy contained within a leaf-labeling of the clades of a tree. First, we introduce some notation. Let $\Sigma$ be an alphabet. Let clade $u$ of tree $T$ be the subtree $T_u$ rooted at node $u$. Let $\delta(u) \subseteq L(T)$ be the subset of leaves that are descendants of node $u$. Let $\ell : L(T) \rightarrow \Sigma$ be a leaf labeling. Given a clade $u$ and leaf-labeling $\ell$, the entropy of a clade with respect to its leaf labels is defined as,

$$H(u, \ell) = -\sum_{s \in [r]} p(s) \log p(s), \tag{Equation 24}$$

where $p(s) = \sum_{v \in \delta(u)} \mathbf{1}(\ell(v) = s)/|\delta(u)|$. The average clade entropy $\overline{H}$ is computed over all clades except the leaves $L(T)$ and the root $r$ as follows,

$$\overline{H}(T, \ell) = \frac{\sum_{u \in \overline{V}} H(u, \ell)}{|\overline{V}|}, \tag{Equation 25}$$

where $\overline{V} = V(T) \backslash (\{r\} \cup L(T))$ is the set of non-trivial clades.

