## [Document S2. Transparent peer review records for Weber et al · Cell Genomics]

Isotype-aware Inference of B cell Clonal Lineage Trees from Single-cell Sequencing Data

Leah L. Weber, Derek Reiman, Mrinmoy S. Roddur, Yuanyuan Qi, Mohammed El-Kebir, Aly Khan

Summary

Initial submission: Received : March 5th 2024

Scientific editor: Laura Zahn and Judith Nicholson

First round of review: Number of reviewers: 3
Revision invited : April 10th 2024
Revision received : June 19th 2024

Second round of review: Number of reviewers: 3
Accepted : 6th August 2024

Data freely available: Yes

Code freely available: Yes

This transparent peer review record is not systematically proofread, type-set, or edited. Special characters, formatting, and equations may fail to render properly. Standard procedural text within the editor's letters has been deleted for the sake of brevity, but all official correspondence specific to the manuscript has been preserved.

Referees' reports, first round of review

Reviewer 1

This study developed an optimized phylogenetic approach, called TRIBAL, to reconstruct more accurate B cell clonal lineage trees based on both SHM and CSR processes using single-cell sequencing data. In particular, the authors firstly solved the B CELL LINEAGE FOREST INFERENCE problem by incorporating the isotype transition probabilities, the order of optimizing SHM and CSR, and the coordinate ascent approach into the algorithm, and solved the MPRT, LP, SET OVER subproblems as well. Comparison of TRIBAL with existing methods through applying them on data from in silico experiments and B cell scRNA-seq data from NP-KLH mice model showed the advantages of using isotype information and tree refinement to minimize phylogenetic uncertainty. Further comparison using three scRNA-seq datasets of age-associated B cells confirmed the great HLP19 likelihoods of TRIBAL and IgPhyML inferred lineage trees and underlined the reduced average isotype clade entropy by TRIBAL. This study makes a reasonable and detailed breakdown of the statistical inference problem at hand, and provides a comprehensive proof. While being statistically rigorous, it also closely corresponds to the biological processes of SHM and CSR in B Cells. This study provides a useful tool for understanding the process of B cell affinity maturation and developing therapeutic antibodies.

The following are several major issues:

- (1) We found that under two parameters of the simulated dataset in direct mode ($n=35$ and $n=65$), the TRIBAL-No Refinement method significantly outperforms TRIBAL in terms of the mean CSR error indicator (Fig. 3d and Fig. S5d). The authors also mentioned in the text that TRIBAL tends to overestimate the number of conversions, while other methods tend to underestimate the number of conversions. But at the same time, the transition probability matrix estimated by TRIBAL is much closer to the simulated ground truth distribution than the estimate of TRIBAL-No Refinement. How should we understand the intuitive contradiction this brings?
- (2) In real data, it is difficult to find many continuous changes of isotype within a clonotype, and it is even difficult to find a clonotype of sufficient size. The NP-KLH series datasets presented by the author also have this feature. Such a

situation may be closer to the direct mode in the simulated data. Combined with the previous question, how should we evaluate the credibility of CSR inference in the real data set?

(3) What is the definition of clonotype in this study? Are SHMs in CDR3 also used by TRIBAL?

The following are a few minor issues:

(1) Some V gene families are easily to be confused during alignment when the antibody sequences accumulate too many SHMs, will TRIBAL correct it during lineage tree construct?

(2) The data part corresponding to the box plot appearing in Figure 3 and the supplementary figures may be non-normal (taking Fig. 3c as an example), and there are too many so-called "outliers". When drawing, the parameter 'whis' may be the default 1.5, and such a default value is given under the assumption that the data approximately follows a normal distribution. To display the data more intuitively, the threshold for determining outliers may need to be adjusted.

(3) We noticed that formula 20 has made a Laplacian smoothing for unobserved isotype transitions, but at the same time did not make the same transformation for observed isotype transitions. Perhaps maintaining consistency is more in line with mathematical logic and habits.

(4) The word "irreversibility" appearing on page 5, line 119 should be "irreversibility".

Reviewer 2

In this study, the authors proposed a method to predict B cell evolution with single cell RNA seq data. Differently from the existing work, the authors developed an algorithm to integrate the SHM and CSR events in the construction of B cell lineages. We noticed the phylogenetic trees built by the method were more well-bedded with lower entropy. The isotype transition probabilities were also inferred to estimate the CSR models by TRIBAL. It can be a good tool in the research of B cell colonization and evolution, but we still come up with some concerns as below.

Since one critical thing when harnessing single cell data is to define the identity for each cell, we would like to know if the cell identity information could be added in the phylogenetic trees. As given the cell fates can be diverse in B cell

evolution, it is better to know not only the isotype transformation information but also the cell type alterations in the B cell developments.

In addition, could the author discuss more about the significance of TRIBAL applied in biology study? We could find the improved performance of the algorithm, but it is hard to interpret if it can bring the enhancement from a perspective of application in biology study.

Reviewer 3

This paper "Isotype-aware Inference of B cell Clonal Lineage Trees from Single-cell Sequencing Data" by Weber et al. describes a new tool is described for the construction of b cell lineage trees based on single-cell DNA sequences. The essence of the TRIBAL method lies in incorporating isotype data into the inference of B cell lineages, inferring isotype transition probabilities, which helps to reduce the phylogenetic uncertainty that arises from considering receptor sequences alone. The tool combines BCR heavy chain and light chain, uses SHM and CSR together to build a lineage tree, which satisfies the minimum SHM consumption and maximum CSR probability within clonotype. Through computer simulation experiments, the authors have demonstrated that TRIBAL can distinguish between direct and sequential switching within B cell populations. Compared to existing methods, it can more accurately reconstruct B cell lineage trees and corresponding ancestral sequences and class switches. Using real-world scRNA-seq datasets, TRIBAL can reproduce expected biological trends in a model of affinity maturation, and it generates lower entropy partitions for the isotype sequences of sequenced B cells.

I think that TRIBAL is based on an elegant and powerful mathematical framework, the article uses a lot of diagrams to illustrate how the tool is built, which is good. Nevertheless, there are several major issues that need to be addressed:

1. Some suggestions for Figure 1 are as follows:

1) Since SHM and CSR do not occur simultaneously during the maturation of B cell, Figure

1(b) is not consistent with the real process of BCR diversity, and can be modified as appropriate

2) Circles of different colors in Figure 1(b) represent different isotypes, respectively

represented by b_0 to b_i (one color corresponds to one isotype b_i); however, b_1

and bk in Figure 1(c) represent isotypes (many colors correspond to an isotype bk), and the representation in Figure 1(c) is consistent with the following text. It is suggested to adjust Figure 1(b) for this conflict phenomenon.

3) There are spelling errors in the figure notes: "leafs" should be changed to "leaves"

2. When using simulated data for tool comparison, have you also compared other common tools in this field, such as MEGA5, BRILIA, AncesTree, GLaMST, IgTreeZ and Abalign? Consider adding an explanation of how the introduction of innovation CSR has improved the effectiveness of the tool.

3. As for TRIBAL algorithm, since it is targeted at single-cell sequencing technology, consider some of the characteristics of single-cell sequencing, it is suggested to further discuss its performance under different conditions, such as changes in data noise, sequence depth and sample diversity, its potential limitations and assumptions in practical application, and the influence these factors may have on the results.

4. In this paper, only the single-cell data of mice were used for real data, but the 10x website provides single-cell data of humans, why the use of mice data? Can the tool be applied to human data? Because the immune environment is different between different species, if it can be applied to human immune data, the response of human vaccines, disease diagnosis is more meaningful.

Authors' response to the first round of review

We thank the reviewers for the favorable comments regarding our work and for taking the time to provide a number of helpful suggestions and thought provoking questions in order to improve the quality of our manuscript. Below is a point-by-point response for each issue identified by the reviewers.

Remarks to the Author:

This paper "Isotype-aware Inference of B cell Clonal Lineage Trees from Single-cell Sequencing Data" by Weber et al. describes a new tool is described for the construction of b cell lineage trees based on single-cell DNA sequences. The essence of the TRIBAL method lies in incorporating isotype data into the inference of B cell lineages, inferring isotype transition probabilities, which helps to reduce the phylogenetic uncertainty that arises from considering receptor sequences alone. The tool combines BCR heavy chain and light chain, uses SHM and CSR together to build a lineage tree, which satisfies the minimum SHM consumption and maximum CSR probability within clonotype. Through computer simulation experiments, the authors have demonstrated that TRIBAL can distinguish between direct and sequential switching within B cell populations. Compared to existing methods, it can more accurately reconstruct B cell lineage trees and corresponding ancestral sequences and class switches. Using real-world scRNA-seq datasets, TRIBAL can reproduce expected biological trends in a model of affinity maturation, and it generates lower entropy partitions for the isotype sequences of sequenced B cells.

I think that TRIBAL is based on an elegant and powerful mathematical framework, the article uses a lot of diagrams to illustrate how the tool is built, which is good.

We thank the reviewer for their careful consideration of our manuscript and are encouraged by their overall positive view of our paper.

Nevertheless, there are several major issues that need to be addressed:

1. Some suggestions for Figure 1 are as follows:

1) Since SHM and CSR do not occur simultaneously during the maturation of B cell, Figure 1(b) is not consistent with the real process of BCR diversity, and can be modified as appropriate

Thank you for the feedback on our figure. We did not realize our depiction might mistakenly imply that SHM and CSR occur simultaneously. We have now separated the previous panel (b) into panels (b) and (c), with one panel for each process. We have also made a number of other changes to further enhance the clarity of the figure.

2) Circles of different colors in Figure 1(b) represent different isotypes, respectively represented by b_{v_0} to b_i (one color corresponds to one isotype b_i); however, b_1 and b_k in Figure 1(c) represent isotypes (many colors correspond to an isotype b_k), and the representation in Figure 1(c) is consistent with the following text. It is suggested to adjust Figure 1(b) for this conflict phenomenon.

As mentioned above, we have made a number of aesthetic changes to Figure 1 to increase its interpretability. One such change is to simplify the isotypes to IgM, IgG, IgE and IgA represented in the figure and a change of isotype color scheme. We hope that our new modified panel (c) representing CSR as well as changes to (d) and (e) increases the clarity and consistency of our example.

3) There are spelling errors in the figure notes: "leafs" should be changed to "leaves"

Thank you for pointing this out. We corrected this and also ran a spell checker to identify any remaining spelling errors.

2. When using simulated data for tool comparison, have you also compared other common tools in this field, such as MEGA5, BRILIA, AncestryTree, GLaMST, IgTreeZ and Abalign?

Thank you for your suggestion to help ensure our simulation analysis is comprehensive. We have carefully reviewed the tools that you suggested and included as many new lineage tree inference methods as possible. Many of the suggested tools appear to be software packages (MEGA5, AncestryTree, IgTreeZ, Abalign) that provide an interface to existing species phylogenetic algorithms, such as maximum likelihood, maximum parsimony methods or IgPhyML. Many of these approaches/methods have already been included in our comparison. However, we have now included neighbor joining (NJ), as this is a commonly used distance based approach in species phylogenetics. Additionally, we opted to include ClonalTree (Abdollahi et al., *BMC Bioinformatics*, 2023) as opposed to the suggested GLaMST for several reasons.

First, GLaMST and ClonalTree both have the same underlying approach in that they seek to find a minimum spanning tree (MST) on a fully connected graph constructed from the sequenced B cells. After finding this MST, both methods then use a heuristic approach to infer the presence of unobserved ancestral nodes. The main difference between these two methods is that since this approach leads to multiple optimal solutions, ClonalTree uses genotype abundance information to prioritize solutions. Although the abundance signal in our simulated data is very weak, i.e., very few duplicated sequences, whether or not to include abundance information is a parameter

of ClonalTree. When not including abundance information, ClonalTree is nearly identical to GlaMST. Therefore we ran ClonalTree in both modes and selected the best solutions. Second, the provided GlaMST implementation only works for windows based computers while ClonalTree is available in python, thus making it far more likely to be a viable competitor to TRIBAL.

Overall, we found that TRIBAL significantly outperformed both NJ and ClonalTree (in both modes) on the accuracy of the tree topology. A fundamental disadvantage of distance based methods, like NJ and ClonalTree is that they do not infer ancestral sequences, which is a highlight of TRIBAL. Also, much like the other methods we included in our comparison (dnaml, dnapars, IgPhyML), distance based methods also do not infer the ancestral isotypes. However, as ClonalTree is an MST based approach it does place sequenced B cells as ancestral to other sequenced B cells while ignoring isotype information. Therefore, we were able to assess if the inferred ClonalTree MST respected the CSR irreversibility constraints. Overall, we found this was not the case and that 92% of the ClonalTree MSTs contained at least one invalid isotype transition. Moreover, we found that 21% of the inferred edges of the ClonalTree MSTs had an invalid isotype transition.

More details on this analysis can be found in our simulation results starting on line 171 of page 6.

Consider adding an explanation of how the introduction of innovation CSR has improved the effectiveness of the tool.

Thank you for pointing out this oversight. We have now added the following text to the first paragraph of the Discussion section on page 14. We hope this clarifies the innovation of including CSR to B cell lineage inference.

“The main innovation of TRIBAL is that the inclusion of isotype data allows us to reduce phylogenetic uncertainty with respect to both the number of optimal solutions and refinement of the evolutionary relationships between B cells. Furthermore, TRIBAL provides isotype transition probabilities and inferred ancestral isotypes, enabling researchers to study CSR dynamics from a single time point and model the interplay between SHM and CSR during the adaptive immune response.”

3. As for TRIBAL algorithm, since it is targeted at single-cell sequencing technology, consider some of the characteristics of single-cell sequencing, it is suggested to further discuss its performance under different conditions, such as changes in data noise,

sequence depth and sample diversity, its potential limitations and assumptions in practical application, and the influence these factors may have on the results.

We appreciate the suggestion. We have sought to address this by applying TRIBAL to three different real-world datasets. In this way, we are utilizing real-world variability in observed sequencing coverage and depth. We have also elaborated in our discussion that our performance is naturally dependent on several upstream tasks, including the performance of tools used for BCR assembly:

“Finally, we note that several upstream steps directly influence our ability to better reconstruct B cell lineage trees. For example, TRIBAL is reliant on the accurate preprocessing of single-cell RNA sequencing data using tools such as Cell Ranger, Dandelion, and Dowser. While these methods have been optimized to minimize the impact of sequencing errors and noise, experimental design choices such as sequencing depth or sample diversity may impact the output of these methods and subsequently the input data to TRIBAL. Future versions of TRIBAL could attempt to identify and mitigate sequencing and preprocessing errors, for example, by allowing inaccurately clonotyped B cells to move between B cell lineage trees.”

As such, if we do not have sufficient coverage for BCR assembly, we cannot assemble a linkage tree. We hope our emphasis on applying this to real-world data provides an appropriate demonstration of the robustness of the tool.

4. In this paper, only the single-cell data of mice were used for real data, but the 10x website provides single-cell data of humans, why the use of mice data? Can the tool be applied to human data? Because the immune environment is different between different species, if it can be applied to human immune data, the response of human vaccines, disease diagnosis is more meaningful.

TRIBAL is agnostic to species and will work for any number of isotypes or isotype ordering as long as the order is specified by the user. We initially chose to only analyze mice model systems because it allowed us to orthogonally validate the accuracy of our inferred B cell lineage trees using known information about affinity maturation. However, we have now included a new analysis on page 12 on a human dataset of a single-cell analysis of individuals vaccinated with the SARS-CoV-2 mRNA-1273 vaccine (Lopes de Assis et al., *Cell Reports*, 2023).

While it is challenging to validate the accuracy of our inferred lineage trees, our analysis highlights the importance of utilizing CSR to reduce phylogenetic uncertainty. Additionally, it demonstrates that TRIBAL infers isotype transition probabilities that

describe similar CSR dynamics to those observed in a previous longitudinal study of the human response to SARS-CoV-2 vaccinations (Ng et al., Nature Methods, 2023).

Reviewers' Comments:

Reviewer #1:

This study developed an optimized phylogenetic approach, called TRIBAL, to reconstruct more accurate B cell clonal lineage trees based on both SHM and CSR processes using single-cell sequencing data. In particular, the authors firstly solved the B CELL LINEAGE FOREST INFERENCE problem by incorporating the isotype transition probabilities, the order of optimizing SHM and CSR, and the coordinate ascent approach into the algorithm, and solved the MPRT, LP, SET OVER subproblems as well. Comparison of TRIBAL with existing methods through applying them on data from in silico experiments and B cell scRNA-seq data from NP-KLH mice model showed the advantages of using isotype information and tree refinement to minimize phylogenetic uncertainty. Further comparison using three scRNA-seq datasets of age-associated B cells confirmed the great HLP19 likelihoods of TRIBAL and IgPhyML inferred lineage trees and underlined the reduced average isotype clade entropy by TRIBAL. This study makes a reasonable and detailed breakdown of the statistical inference problem at hand, and provides a comprehensive proof. While being statistically rigorous, it also closely corresponds to the biological processes of SHM and CSR in B Cells. This study provides a useful tool for understanding the process of B cell affinity maturation and developing therapeutic antibodies.

We thank the reviewer for their careful consideration of our manuscript and are encouraged by their overall positive view of our paper.

The following are several major issues:

(1) We found that under two parameters of the simulated dataset in direct mode ($n=35$ and $n=65$), the TRIBAL-No Refinement method significantly outperforms TRIBAL in terms of the mean CSR error indicator (Fig. 3d and Fig. S5d). The authors also mentioned in the text that TRIBAL tends to overestimate the number of conversions, while other methods tend to underestimate the number of conversions. But at the same time, the transition probability matrix estimated by TRIBAL is much closer to the simulated ground truth distribution than the estimate of TRIBAL-No Refinement. How should we understand the intuitive contradiction this brings?

This is an excellent question. These overestimations for a single clonotype under direct switching tend to arise due to using the maximum likelihood estimates as opposed to considering the marginal distribution at each ancestral node. Considering this marginal

distribution when inferring ancestral isotypes is a much more complicated problem than the MPTR problem. We now write the following in line 260 on page 9:

“This slight overestimation is likely due to utilizing the maximum likelihood estimates of the inferred ancestral isotypes, as opposed to considering the marginal distribution of ancestral isotype states for each node. In other words, for any given clonotype, it is difficult to infer if the unobserved ancestral isotypes underwent direct or sequential class-switching but given multiple clonotypes, TRIBAL is able to more accurately tease out these relative frequencies in class-switching than TRIBAL-NR (Fig. 3a).”

(2) In real data, it is difficult to find many continuous changes of isotype within a clonotype, and it is even difficult to find a clonotype of sufficient size. The NP-KLH series datasets presented by the author also have this feature. Such a situation may be closer to the direct mode in the simulated data.

One advantage of our method is that clonotypes with fewer B cells can borrow strength from those few clonotypes that have a larger number of cells, assuming they share the same transition parameters. In the event that all clonotypes have only a very small number of cells, TRIBAL, will be less effective at inferring isotype transition probabilities and will be equivalent to using parsimony based methods.

We also note the emergence of Parse Biosciences, whose new technology offers high throughput single-cell sequencing of ~1 million B cells with BCRs and whole transcriptome. As throughput increases, there is much greater opportunity to observe more continuous changes of isotype within a clonotype. In our newly added analysis of human data in response to SARs-CoV-2 vaccine (see our response to question 4 above), we do note that 35% of the clonotypes we evaluated contained two or more isotypes.

Combined with the previous question, how should we evaluate the credibility of CSR inference in the real data set?

This is a great question. First, we hope that validating CSR inference with known ground truth via simulations increases the credibility of our method. But of course, it is very difficult to fully capture the complexities of B cell evolution via simulations. One way that we can validate CSR inference on real data is to compare the results of TRIBAL with orthogonal methods. For example, scICSR (Ng et al., *Nature Communications*, 2023) models CSR dynamics using germline sterile transcripts from longitudinal data. While, TRIBAL does not require longitudinal data, we were able to infer similar CSR dynamics in response to SARS-CoV-2 mRNA vaccine using an independent dataset.

Such consistency between methods and across experimental studies is helpful for increasing the credibility of CSR inference for both methods. See the paragraph starting on line 396 for comparison of these CSR trends across methods and datasets.

(3) What is the definition of clonotype in this study? Are SHMs in CDR3 also used by TRIBAL?

The formal definition of a clonotype from the perspective of TRIBAL is a set of cells that all descend from the same naive BCR. In practice, we make use of Dandelion to reassign the BCR V gene allele calls and then cluster the B cells into clonotypes. Dandelion specifies clonotypes using “the following ordered criteria for both heavy and light chain contigs as follows: (i) identical V and J genes usage; (ii) identical junctional CDR3 amino acid length and (iii) CDR3 sequence similarity with the default setting for BCRs set to 85% amino acid sequence similarity based on Hamming distance. Network analysis is then used to assign clusters. (Suo et al., *Nature Biotechnology*, 2024)”. Additionally, SHMs in CDR3 are also used by TRIBAL and are the basis for our alignment step and informs our SHM lineage reconstruction.

The following are a few minor issues:

(1) Some V gene families are easily to be confused during alignment when the antibody sequences accumulate too many SHMs, will TRIBAL correct it during lineage tree construct?

This is an excellent question. We understand this limitation and plan to incorporate the opportunity for B cells to move between clonotypes in future versions of TRIBAL. However, TRIBAL currently does not yet correct for errors in clonotyping and inaccuracies in clonotyping may impact downstream analysis. This is the reason we recommend making use of state of the art methods such as Dandelion for preprocessing and do not yet attempt to jointly solve the problem of clonotyping and B cell lineage inference. We note that the last paragraph of our discussion includes the future direction that “*TRIBAL could be extended to allow for correction of inaccurately clonotyped B cells.*”

(2) The data part corresponding to the box plot appearing in Figure 3 and the supplementary figures may be non-normal (taking Fig. 3c as an example), and there are too many so-called "outliers". When drawing, the parameter 'whis' may be the default 1.5, and such a default value is given under the assumption that the data approximately follows a normal distribution. To display the data more intuitively, the threshold for determining outliers may need to be adjusted.

Thank you for the helpful suggestion. We have experimented with this parameter and chose to use a “whis” parameter setting of 5 in order to more intuitively display the data.

(3) We noticed that formula 20 has made a Laplacian smoothing for unobserved isotype transitions, but at the same time did not make the same transformation for observed isotype transitions. Perhaps maintaining consistency is more in line with mathematical logic and habits.

We apologize for this confusion. We do apply this smoothing to estimate all isotype transition probabilities and not simply unobserved isotype transitions. Upon reviewing the text, we realized the previous wording of this sentence was confusing and have modified it as follows:

“Lastly, we apply a pseudocount of 1 to all isotype transition probabilities $p_{s,t}$ where $s < t$, in order to account for the potential of any unobserved transitions.”

(4) The word “irreversibility” appearing on page 5, line 119 should be “irreversibility”.

Thanks. We have run a spell checker and corrected this error.

Reviewer #2: In this study, the authors proposed a method to predict B cell evolution with single cell RNA seq data. Differently from the existing work, the authors developed an algorithm to integrate the SHM and CSR events in the construction of B cell lineages. We noticed the phylogenetic trees built by the method were more well-bedded with lower entropy. The isotype transition probabilities were also inferred to estimate the CSR models by TRIBAL. It can be a good tool in the research of B cell colonization and evolution, but we still come up with some concerns as below.

Thank you for recognizing the positive results of our work and the contribution of our method to the existing literature. We hope that we are able to address your additional concerns below.

Since one critical thing when harnessing single cell data is to define the identity for each cell, we would like to know if the cell identity information could be added in the phylogenetic trees. As given the cell fates can be diverse in B cell evolution, it is better to know not only the isotype transformation information but also the cell type alterations in the B cell developments.

This is a great suggestion and something that we have been thinking a lot about lately. It is indeed possible to extend TRIBAL in the future to jointly model SHM, CSR and cell

fate utilizing transcriptome data. We have added the following sentence to the last paragraph of our discussion in order to highlight this future possible extension of TRIBAL.

“Fifth, TRIBAL could also be extended to jointly model SHM, CSR, and B cell states (e.g., naive, memory) derived from the sequenced transcriptome to provide a more comprehensive reconstruction of B cell evolution during the adaptive immune response.”

In addition, could the author discuss more about the significance of TRIBAL applied in biology study? We could find the improved performance of the algorithm, but it is hard to interpret if it can bring the enhancement from a perspective of application in biology study.

We apologize for our oversight in making the anticipated contribution of TRIBAL to biological research explicit. We have now added the following paragraph to our discussion.

“Additionally, advancements in single-cell sequencing technologies promise increased efficiency in cell capture and high-throughput profiling. For instance, Parse Biosciences recently sequenced one million B cells from multiple patient cohorts (<https://www.parsebiosciences.com/datasets/bcr-sequencing-of-1-million-healthy-and-diseased-samples-in-a-single-experiment/>). This increase in cell numbers presents new computing challenges. However, as TRIBAL is the only method to model both SHM and CSR, it is well-suited to help researchers understand the relationship between SHM and CSR and to elucidate CSR dynamics within and across different disease cohorts at large scale. Ultimately, we hope that TRIBAL will enhance our knowledge of basic biology and aid in designing vaccines and treatments.”

We thank the reviewers for the favorable comments regarding our work and for taking the time to provide a number of helpful suggestions and thought provoking questions in order to improve the quality of our manuscript. Below is a point-by-point response for each issue identified by the reviewers.

Remarks to the Author:

This paper "Isotype-aware Inference of B cell Clonal Lineage Trees from Single-cell Sequencing Data" by Weber et al. describes a new tool is described for the construction of b cell lineage trees based on single-cell DNA sequences. The essence of the TRIBAL method lies in incorporating isotype data into the inference of B cell lineages, inferring isotype transition probabilities, which helps to reduce the phylogenetic uncertainty that arises from considering receptor sequences alone. The tool combines BCR heavy chain and light chain, uses SHM and CSR together to build a lineage tree, which satisfies the minimum SHM consumption and maximum CSR probability within clonotype. Through computer simulation experiments, the authors have demonstrated that TRIBAL can distinguish between direct and sequential switching within B cell populations. Compared to existing methods, it can more accurately reconstruct B cell lineage trees and corresponding ancestral sequences and class switches. Using real-world scRNA-seq datasets, TRIBAL can reproduce expected biological trends in a model of affinity maturation, and it generates lower entropy partitions for the isotype sequences of sequenced B cells.

I think that TRIBAL is based on an elegant and powerful mathematical framework, the article uses a lot of diagrams to illustrate how the tool is built, which is good.

We thank the reviewer for their careful consideration of our manuscript and are encouraged by their overall positive view of our paper.

Nevertheless, there are several major issues that need to be addressed:

1. Some suggestions for Figure 1 are as follows:

1) Since SHM and CSR do not occur simultaneously during the maturation of B cell, Figure 1(b) is not consistent with the real process of BCR diversity, and can be modified as appropriate

Thank you for the feedback on our figure. We did not realize our depiction might mistakenly imply that SHM and CSR occur simultaneously. We have now separated the previous panel (b) into panels (b) and (c), with one panel for each process. We have also made a number of other changes to further enhance the clarity of the figure.

2) Circles of different colors in Figure 1(b) represent different isotypes, respectively represented by b_{v_0} to b_i (one color corresponds to one isotype b_i); however, b_1 and b_k in Figure 1(c) represent isotypes (many colors correspond to an isotype b_k), and the representation in Figure 1(c) is consistent with the following text. It is suggested to adjust Figure 1(b) for this conflict phenomenon.

As mentioned above, we have made a number of aesthetic changes to Figure 1 to increase its interpretability. One such change is to simplify the isotypes to IgM, IgG, IgE and IgA represented in the figure and a change of isotype color scheme. We hope that our new modified panel (c) representing CSR as well as changes to (d) and (e) increases the clarity and consistency of our example.

3) There are spelling errors in the figure notes: "leafs" should be changed to "leaves"

Thank you for pointing this out. We corrected this and also ran a spell checker to identify any remaining spelling errors.

2. When using simulated data for tool comparison, have you also compared other common tools in this field, such as MEGA5, BRILIA, Ancestree, GLaMST, IgTreeZ and Abalign?

Thank you for your suggestion to help ensure our simulation analysis is comprehensive. We have carefully reviewed the tools that you suggested and included as many new lineage tree inference methods as possible. Many of the suggested tools appear to be software packages (MEGA5, Ancestree, IgTreeZ, Abalign) that provide an interface to existing species phylogenetic algorithms, such as maximum likelihood, maximum parsimony methods or IgPhyML. Many of these approaches/methods have already been included in our comparison. However, we have now included neighbor joining (NJ), as this is a commonly used distance based approach in species phylogenetics. Additionally, we opted to include ClonalTree (Abdollahi et al., *BMC Bioinformatics*, 2023) as opposed to the suggested GLaMST for several reasons.

First, GLaMST and ClonalTree both have the same underlying approach in that they seek to find a minimum spanning tree (MST) on a fully connected graph constructed from the sequenced B cells. After finding this MST, both methods then use a heuristic approach to infer the presence of unobserved ancestral nodes. The main difference between these two methods is that since this approach leads to multiple optimal solutions, ClonalTree uses genotype abundance information to prioritize solutions. Although the abundance signal in our simulated data is very weak, i.e., very few duplicated sequences, whether or not to include abundance information is a parameter

of ClonalTree. When not including abundance information, ClonalTree is nearly identical to GlaMST. Therefore we ran ClonalTree in both modes and selected the best solutions. Second, the provided GlaMST implementation only works for windows based computers while ClonalTree is available in python, thus making it far more likely to be a viable competitor to TRIBAL.

Overall, we found that TRIBAL significantly outperformed both NJ and ClonalTree (in both modes) on the accuracy of the tree topology. A fundamental disadvantage of distance based methods, like NJ and ClonalTree is that they do not infer ancestral sequences, which is a highlight of TRIBAL. Also, much like the other methods we included in our comparison (dnaml, dnapars, IgPhyML), distance based methods also do not infer the ancestral isotypes. However, as ClonalTree is an MST based approach it does place sequenced B cells as ancestral to other sequenced B cells while ignoring isotype information. Therefore, we were able to assess if the inferred ClonalTree MST respected the CSR irreversibility constraints. Overall, we found this was not the case and that 92% of the ClonalTree MSTs contained at least one invalid isotype transition. Moreover, we found that 21% of the inferred edges of the ClonalTree MSTs had an invalid isotype transition.

More details on this analysis can be found in our simulation results starting on line 171 of page 6.

Consider adding an explanation of how the introduction of innovation CSR has improved the effectiveness of the tool.

Thank you for pointing out this oversight. We have now added the following text to the first paragraph of the Discussion section on page 14. We hope this clarifies the innovation of including CSR to B cell lineage inference.

“The main innovation of TRIBAL is that the inclusion of isotype data allows us to reduce phylogenetic uncertainty with respect to both the number of optimal solutions and refinement of the evolutionary relationships between B cells. Furthermore, TRIBAL provides isotype transition probabilities and inferred ancestral isotypes, enabling researchers to study CSR dynamics from a single time point and model the interplay between SHM and CSR during the adaptive immune response.”

3. As for TRIBAL algorithm, since it is targeted at single-cell sequencing technology, consider some of the characteristics of single-cell sequencing, it is suggested to further discuss its performance under different conditions, such as changes in data noise,

sequence depth and sample diversity, its potential limitations and assumptions in practical application, and the influence these factors may have on the results.

We appreciate the suggestion. We have sought to address this by applying TRIBAL to three different real-world datasets. In this way, we are utilizing real-world variability in observed sequencing coverage and depth. We have also elaborated in our discussion that our performance is naturally dependent on several upstream tasks, including the performance of tools used for BCR assembly:

“Finally, we note that several upstream steps directly influence our ability to better reconstruct B cell lineage trees. For example, TRIBAL is reliant on the accurate preprocessing of single-cell RNA sequencing data using tools such as Cell Ranger, Dandelion, and Dowser. While these methods have been optimized to minimize the impact of sequencing errors and noise, experimental design choices such as sequencing depth or sample diversity may impact the output of these methods and subsequently the input data to TRIBAL. Future versions of TRIBAL could attempt to identify and mitigate sequencing and preprocessing errors, for example, by allowing inaccurately clonotyped B cells to move between B cell lineage trees.”

As such, if we do not have sufficient coverage for BCR assembly, we cannot assemble a linkage tree. We hope our emphasis on applying this to real-world data provides an appropriate demonstration of the robustness of the tool.

4. In this paper, only the single-cell data of mice were used for real data, but the 10x website provides single-cell data of humans, why the use of mice data? Can the tool be applied to human data? Because the immune environment is different between different species, if it can be applied to human immune data, the response of human vaccines, disease diagnosis is more meaningful.

TRIBAL is agnostic to species and will work for any number of isotypes or isotype ordering as long as the order is specified by the user. We initially chose to only analyze mice model systems because it allowed us to orthogonally validate the accuracy of our inferred B cell lineage trees using known information about affinity maturation. However, we have now included a new analysis on page 12 on a human dataset of a single-cell analysis of individuals vaccinated with the SARS-CoV-2 mRNA-1273 vaccine (Lopes de Assis et al., *Cell Reports*, 2023).

While it is challenging to validate the accuracy of our inferred lineage trees, our analysis highlights the importance of utilizing CSR to reduce phylogenetic uncertainty. Additionally, it demonstrates that TRIBAL infers isotype transition probabilities that

describe similar CSR dynamics to those observed in a previous longitudinal study of the human response to SARS-CoV-2 vaccinations (Ng et al., Nature Methods, 2023).

Reviewers' Comments:

Reviewer #1:

This study developed an optimized phylogenetic approach, called TRIBAL, to reconstruct more accurate B cell clonal lineage trees based on both SHM and CSR processes using single-cell sequencing data. In particular, the authors firstly solved the B CELL LINEAGE FOREST INFERENCE problem by incorporating the isotype transition probabilities, the order of optimizing SHM and CSR, and the coordinate ascent approach into the algorithm, and solved the MPRT, LP, SET OVER subproblems as well. Comparison of TRIBAL with existing methods through applying them on data from in silico experiments and B cell scRNA-seq data from NP-KLH mice model showed the advantages of using isotype information and tree refinement to minimize phylogenetic uncertainty. Further comparison using three scRNA-seq datasets of age-associated B cells confirmed the great HLP19 likelihoods of TRIBAL and IgPhyML inferred lineage trees and underlined the reduced average isotype clade entropy by TRIBAL. This study makes a reasonable and detailed breakdown of the statistical inference problem at hand, and provides a comprehensive proof. While being statistically rigorous, it also closely corresponds to the biological processes of SHM and CSR in B Cells. This study provides a useful tool for understanding the process of B cell affinity maturation and developing therapeutic antibodies.

We thank the reviewer for their careful consideration of our manuscript and are encouraged by their overall positive view of our paper.

The following are several major issues:

(1) We found that under two parameters of the simulated dataset in direct mode ($n=35$ and $n=65$), the TRIBAL-No Refinement method significantly outperforms TRIBAL in terms of the mean CSR error indicator (Fig. 3d and Fig. S5d). The authors also mentioned in the text that TRIBAL tends to overestimate the number of conversions, while other methods tend to underestimate the number of conversions. But at the same time, the transition probability matrix estimated by TRIBAL is much closer to the simulated ground truth distribution than the estimate of TRIBAL-No Refinement. How should we understand the intuitive contradiction this brings?

This is an excellent question. These overestimations for a single clonotype under direct switching tend to arise due to using the maximum likelihood estimates as opposed to considering the marginal distribution at each ancestral node. Considering this marginal

distribution when inferring ancestral isotypes is a much more complicated problem than the MPTR problem. We now write the following in line 260 on page 9:

“This slight overestimation is likely due to utilizing the maximum likelihood estimates of the inferred ancestral isotypes, as opposed to considering the marginal distribution of ancestral isotype states for each node. In other words, for any given clonotype, it is difficult to infer if the unobserved ancestral isotypes underwent direct or sequential class-switching but given multiple clonotypes, TRIBAL is able to more accurately tease out these relative frequencies in class-switching than TRIBAL-NR (Fig. 3a).”

(2) In real data, it is difficult to find many continuous changes of isotype within a clonotype, and it is even difficult to find a clonotype of sufficient size. The NP-KLH series datasets presented by the author also have this feature. Such a situation may be closer to the direct mode in the simulated data.

One advantage of our method is that clonotypes with fewer B cells can borrow strength from those few clonotypes that have a larger number of cells, assuming they share the same transition parameters. In the event that all clonotypes have only a very small number of cells, TRIBAL, will be less effective at inferring isotype transition probabilities and will be equivalent to using parsimony based methods.

We also note the emergence of Parse Biosciences, whose new technology offers high throughput single-cell sequencing of ~1 million B cells with BCRs and whole transcriptome. As throughput increases, there is much greater opportunity to observe more continuous changes of isotype within a clonotype. In our newly added analysis of human data in response to SARs-CoV-2 vaccine (see our response to question 4 above), we do note that 35% of the clonotypes we evaluated contained two or more isotypes.

Combined with the previous question, how should we evaluate the credibility of CSR inference in the real data set?

This is a great question. First, we hope that validating CSR inference with known ground truth via simulations increases the credibility of our method. But of course, it is very difficult to fully capture the complexities of B cell evolution via simulations. One way that we can validate CSR inference on real data is to compare the results of TRIBAL with orthogonal methods. For example, scICSR (Ng et al., *Nature Communications*, 2023) models CSR dynamics using germline sterile transcripts from longitudinal data. While, TRIBAL does not require longitudinal data, we were able to infer similar CSR dynamics in response to SARS-CoV-2 mRNA vaccine using an independent dataset.

Such consistency between methods and across experimental studies is helpful for increasing the credibility of CSR inference for both methods. See the paragraph starting on line 396 for comparison of these CSR trends across methods and datasets.

(3) What is the definition of clonotype in this study? Are SHMs in CDR3 also used by TRIBAL?

The formal definition of a clonotype from the perspective of TRIBAL is a set of cells that all descend from the same naive BCR. In practice, we make use of Dandelion to reassign the BCR V gene allele calls and then cluster the B cells into clonotypes. Dandelion specifies clonotypes using “the following ordered criteria for both heavy and light chain contigs as follows: (i) identical V and J genes usage; (ii) identical junctional CDR3 amino acid length and (iii) CDR3 sequence similarity with the default setting for BCRs set to 85% amino acid sequence similarity based on Hamming distance. Network analysis is then used to assign clusters. (Suo et al., *Nature Biotechnology*, 2024)”. Additionally, SHMs in CDR3 are also used by TRIBAL and are the basis for our alignment step and informs our SHM lineage reconstruction.

The following are a few minor issues:

(1) Some V gene families are easily to be confused during alignment when the antibody sequences accumulate too many SHMs, will TRIBAL correct it during lineage tree construct?

This is an excellent question. We understand this limitation and plan to incorporate the opportunity for B cells to move between clonotypes in future versions of TRIBAL. However, TRIBAL currently does not yet correct for errors in clonotyping and inaccuracies in clonotyping may impact downstream analysis. This is the reason we recommend making use of state of the art methods such as Dandelion for preprocessing and do not yet attempt to jointly solve the problem of clonotyping and B cell lineage inference. We note that the last paragraph of our discussion includes the future direction that “*TRIBAL could be extended to allow for correction of inaccurately clonotyped B cells.*”

(2) The data part corresponding to the box plot appearing in Figure 3 and the supplementary figures may be non-normal (taking Fig. 3c as an example), and there are too many so-called "outliers". When drawing, the parameter 'whis' may be the default 1.5, and such a default value is given under the assumption that the data approximately follows a normal distribution. To display the data more intuitively, the threshold for determining outliers may need to be adjusted.

Thank you for the helpful suggestion. We have experimented with this parameter and chose to use a “whis” parameter setting of 5 in order to more intuitively display the data.

(3) We noticed that formula 20 has made a Laplacian smoothing for unobserved isotype transitions, but at the same time did not make the same transformation for observed isotype transitions. Perhaps maintaining consistency is more in line with mathematical logic and habits.

We apologize for this confusion. We do apply this smoothing to estimate all isotype transition probabilities and not simply unobserved isotype transitions. Upon reviewing the text, we realized the previous wording of this sentence was confusing and have modified it as follows:

“Lastly, we apply a pseudocount of 1 to all isotype transition probabilities $p_{s,t}$ where $s < t$, in order to account for the potential of any unobserved transitions.”

(4) The word “irreversibility” appearing on page 5, line 119 should be “irreversibility”.

Thanks. We have run a spell checker and corrected this error.

Reviewer #2: In this study, the authors proposed a method to predict B cell evolution with single cell RNA seq data. Differently from the existing work, the authors developed an algorithm to integrate the SHM and CSR events in the construction of B cell lineages. We noticed the phylogenetic trees built by the method were more well-bedded with lower entropy. The isotype transition probabilities were also inferred to estimate the CSR models by TRIBAL. It can be a good tool in the research of B cell colonization and evolution, but we still come up with some concerns as below.

Thank you for recognizing the positive results of our work and the contribution of our method to the existing literature. We hope that we are able to address your additional concerns below.

Since one critical thing when harnessing single cell data is to define the identity for each cell, we would like to know if the cell identity information could be added in the phylogenetic trees. As given the cell fates can be diverse in B cell evolution, it is better to know not only the isotype transformation information but also the cell type alterations in the B cell developments.

This is a great suggestion and something that we have been thinking a lot about lately. It is indeed possible to extend TRIBAL in the future to jointly model SHM, CSR and cell

fate utilizing transcriptome data. We have added the following sentence to the last paragraph of our discussion in order to highlight this future possible extension of TRIBAL.

“Fifth, TRIBAL could also be extended to jointly model SHM, CSR, and B cell states (e.g., naive, memory) derived from the sequenced transcriptome to provide a more comprehensive reconstruction of B cell evolution during the adaptive immune response.”

In addition, could the author discuss more about the significance of TRIBAL applied in biology study? We could find the improved performance of the algorithm, but it is hard to interpret if it can bring the enhancement from a perspective of application in biology study.

We apologize for our oversight in making the anticipated contribution of TRIBAL to biological research explicit. We have now added the following paragraph to our discussion.

“Additionally, advancements in single-cell sequencing technologies promise increased efficiency in cell capture and high-throughput profiling. For instance, Parse Biosciences recently sequenced one million B cells from multiple patient cohorts (<https://www.parsebiosciences.com/datasets/bcr-sequencing-of-1-million-healthy-and-diseased-samples-in-a-single-experiment/>). This increase in cell numbers presents new computing challenges. However, as TRIBAL is the only method to model both SHM and CSR, it is well-suited to help researchers understand the relationship between SHM and CSR and to elucidate CSR dynamics within and across different disease cohorts at large scale. Ultimately, we hope that TRIBAL will enhance our knowledge of basic biology and aid in designing vaccines and treatments.”

Referees' report, second round of review

Reviewer 1

The authors have fully addressed my previous concern.

Reviewer 2

This revised manuscript has addressed my concerns, I don't have additional comments.

Reviewer 3

All my comments have been properly addressed.

Authors' response to the second round of review